# Measurement Report: Optical and structural properties of atmospheric water-soluble organic carbon in China: Insights from multi-site spectroscopic measurements

Haibiao Chen[1,2], Caiqing Yan[1], Liubin Huang[1], Lin Du[1,3], Yang Yue[3], Xinfeng Wang[1], Qingcai Chen[4], Mingjie Xie[2], Junwen Liu[5], Fengwen Wang[6], Shuhong Fang[7], Qiaoyun Yang[8], Hongya Niu[9], Mei Zheng[10], Yan Wu[3], Likun Xue[1]

[1]Qingdao Key Laboratory for Prevention and Control of Atmospheric Pollution in Coastal Cities, Environment Research Institute, Shandong University, Qingdao, 266237, China
[2]Collaborative Innovation Center of Atmospheric Environment and Equipment Technology, Jiangsu Key Laboratory of Atmospheric Environment Monitoring and Pollution Control (AEMPC), Nanjing University of Information Science & Technology, Nanjing, 210044, China
[3]School of Environmental Science and Engineering, Shandong University, Qingdao, 266237, China
[4]School of Environmental Science and Engineering, Shaanxi University of Science and Technology, Xi'an, 710021, China
[5]Institute for Environmental and Climate Research, Jinan University, Guangzhou, 511443, China
[6]State Key Laboratory of Coal Mine Disaster Dynamics and Control, Department of Environmental Science, Chongqing University, Chongqing, 400030, China
[7]College of Resources and Environment, Chengdu University of Information Technology, Chengdu, 610225, China
[8]Department of Occupational and Environmental Health, School of Public Health, Tianjin Medical University, Tianjin, 300070, China
[9]Key Laboratory of Resource Exploration Research of Hebei Province, Hebei University of Engineering, Handan, 056038, China
[10]SKL-ESPC, College of Environmental Sciences and Engineering, and Center for Environment and Health, Peking University, Beijing, 100871, China

*Correspondence to*: Caiqing Yan (cyan0325@sdu.edu.cn)

**Abstract.** To understand the spatial variation of optical and structural properties of water-soluble brown carbon and its influencing factors in China, the light absorption, fluorescence, and Fourier transform infrared (FTIR) spectrum of water-soluble organic carbon (WSOC) in different regions of China are measured following the same analytical methods. The average light absorption coefficients and mass absorption efficiencies of WSOC at 365 nm ($Abs_{365}$ and $MAE_{365}$) rank from high to low as northwest China > southwest China > north China > east China > regional site, with higher values in Northern China than Southern China and regional sites, and higher values in inland areas than coastal areas. The light absorption factors resolved by light absorption spectra-based positive matrix factorization model, and the abundance of aromatic O-H and C=C functional groups determined by FTIR, both indicate that aromatic compounds are significant light-absorbing substances in WSOC and have a significant impact on fluorophores. Multiple linear regression analysis shows that the fluorophores identified by fluorescence spectra combined with parallel factor analysis contribute to about 62-93% of the WSOC light absorption at all sites, in which humic-like substance (HULIS) contributes the most, especially highly-oxygenated HULIS (29%-50%) with long emission wavelengths. Combustion source emissions and atmospheric chemical processes have significant impacts on

the WSOC light absorption at some sites. Moreover, relative humidity (RH) can also affect $MAE_{365}$ of WSOC with $MAE_{365}$ values decreasing with the increase of RH when RH < 60% and remaining relatively unchanged when RH > 60%. Taken together, this study promotes a better understanding of the spatial heterogeneity of optical and structural properties of WSOC and their influencing factors in China.

## 1 Introduction

Brown carbon (BrC) is an important type of carbonaceous aerosol that absorbs light over the ultraviolet and visible (UV-Vis) range and exhibits a strong wavelength dependence (Andreae and Gelencsér, 2006). BrC has substantial effects on atmospheric radiative forcing and regional climate due to its strong light absorption capacity in the UV-Vis range and has attracted widespread attention in recent years (Laskin et al., 2015; Wang et al., 2022b). It has been simulated that BrC contributes up to 72% of the total light absorption of aerosols at 370 nm globally and the global direct radiative effect of BrC (+0.048 $W \cdot m^{-2}$) is about 30% of black carbon (+0.17 $W \cdot m^{-2}$) (Wang et al., 2018). However, the lack of field measurements limits the model's ability to extend to global simulations of BrC. The spatiotemporal variation of optical properties of BrC is one of the key factors leading to the uncertainty in the radiation assessment of organic aerosols. A comprehensive and clear understanding of optical properties of BrC in different regions is essential to accurately assess the atmospheric radiative forcing of aerosols. Solvent-soluble organic carbon (e.g., water-soluble organic carbon, WSOC; methanol-soluble organic carbon, MSOC) is often used to act as a substitute of BrC. In particular, light absorption of WSOC has been extensively studied, due to its widespread presence and high atmospheric abundance in the atmosphere, as well as mature extraction methods, although some previous studies have indicated that water-insoluble OC (WISOC) contains more light-absorbing BrC (Cao et al., 2021; Chen et al., 2024; Cheng et al., 2016; Yan et al., 2017). Absorption and fluorescence spectroscopy are two of the most widely used methods to reveal optical properties of WSOC (Wang et al., 2022b; Wu et al., 2021). By light-absorption spectroscopy analysis, light absorption characteristics and capabilities of WSOC from different sources or environments are usually characterized by the absorption coefficient or mass absorption efficiency of a specific wavelength over the range of 360-370 nm (average 365 nm) (Hecobian et al., 2010). And the direct radiative forcing of WSOC can be further estimated by simplified radiative forcing models combined with the measured absorption coefficient. The fluorescence spectra obtained by fluorescence spectroscopy measurement could be used to characterize fluorescence fingerprints of WSOC, and provide source-related information of WSOC according to fluorescent indices. In addition, the parallel factor analysis (PARAFAC) could be used to analyze fluorescence spectra and help to infer the chemical and structural characteristics of WSOC chromophores (Chen et al., 2020; Wu et al., 2021).

Spectroscopy-based studies conducted in different countries or regions all over the world show that there are significant spatiotemporal differences in the light absorption characteristics of WSOC. For example, the mass absorption efficiency of WSOC is generally higher in Asia than in North America and Europe, and higher in winter compared to summer (Hecobian et al., 2010; Kirillova et al., 2014; Teich et al., 2017; Wang et al., 2022a). Furthermore, light absorption of WSOC can

also vary from location to location even within the same region (Cao et al., 2024b). However, the current understanding of spatial differences in light absorption and fluorescence characteristics of WSOC is mainly based on the results from different laboratories with different extraction and analysis methods. This may affect the interpretation and accuracy of comparison results due to differences in the analysis methods. For example, it has been shown that the size of the syringe filter could lead to differences in the solvent-soluble BrC light absorption measurement results (Zhang et al., 2022a). Therefore, it will be more convincing and necessary to use a uniform method to compare the optical properties of WSOC in different regions.

Previous studies have suggested that there may be a link between the light absorption and fluorescence components of BrC (Chen et al., 2019; Niu et al., 2022; Tang et al., 2021), as the necessary condition for an organic compound to produce a fluorescent signal is that it absorbs light. This means that fluorescent substances must be able to absorb light, and substances that can absorb light are not necessarily fluorescent. However, studies on the relationship between light absorption and fluorescence of BrC are still very limited, and the contribution of fluorescent components identified by the commonly used three-dimensional fluorescence spectroscopy to the light absorption of BrC has not been well quantified. Additionally, Fourier transform infrared (FTIR) has been used for the structural characterization of BrC as a non-destructive spectral analysis method that can provide the functional group structure information of compounds (Dey and Sarkar, 2024). The addition of FTIR spectroscopy analysis helps to better understand the structural features that affect the optical properties of BrC. However, in previous spectroscopy-based studies, the results of different spectral methods are often discussed separately, without in-depth discussion of the relationship between different spectra, which limits the full and comprehensive prediction of the optical and structural characteristics of BrC.

In this study, $PM_{2.5}$ is collected from ten sites in different regions of China. The mass concentration, light absorption and fluorescent spectra, and functional group structures of WSOC (a substitute of water-soluble BrC) at the ten sites are analyzed using a unified method. The objectives of this study include: (1) to explore the spatial heterogeneity of optical properties of WSOC in different regions of China and its influencing factors, (2) to reveal the relationship between light absorption, fluorescence and functional group structure of WSOC, and (3) to quantify the contribution of fluorescent chromophores to the light absorption of WSOC on the basis of multi-site spectral datasets. Furthermore, this study can also provide reference for the future study of the light absorption properties and structural composition of WISOC.

## 2 Material and methods

### 2.1 Ambient $PM_{2.5}$ sample collection

$PM_{2.5}$ samples are collected at eight urban sites and two regional sites in China during the late November and January of 2019-2020 (see Figure S1 and Table S1). The eight cities are distributed in different administrative regions of China, including Tianjin (TJ) and Handan (HD) in north China, Qingdao (QD), Nanjing (NJ) and Shanghai (SH) in east China, Xi'an (XA) in northwest China, Chengdu (CD) and Chongqing (CQ) in southwest China. These cities are representative of their respective

regional economies and cultures, with discrepancies in energy structure, geographical and climatic characteristics. The other two sites, Mt. Tai (TS) and Heshan (HS), are taken as regional sites in this study. TS (1534 m a.s.l.) locates at the Taishan National Reference Climatological Station at the summit of Mt. Tai and in the middle of the North China Plain, which is less affected by anthropogenic emissions and has been widely used as a sampling site for researches on regional atmospheric pollution and atmospheric chemistry in Northern China (Chen et al., 2022; Jiang et al., 2020). The HS site locates at the Atmospheric Environmental Monitoring Super-station in Guangdong, China and downwind of the Pearl River Delta (PRD), which is mainly surrounded by farmland protection areas and forest land with no obvious industrial or urban traffic pollution sources in the vicinity, and has been used as a representative regional receptor site for the PRD region (He et al., 2019; Xu et al., 2022). In this study, daytime, nighttime or daily $PM_{2.5}$ samples were collected using medium- or high-volume samplers at different sites with a sampling duration of 11 h-24 h for each sample. More detailed sampling information and sample sizes are summarized in Table S1 in the Supplement. It is worth noting that in this study, the daily average of the parameters measured at each site is used for subsequent summary and comparison. Before sampling, all quartz filters used for sample collection are prebaked at 550 ℃ for 5.5 h to remove potential organic matter. After sampling, all samples are wrapped in prebaked aluminum foils and stored in the refrigerator under -20 ℃ until further analysis.

## 2.2 Carbonaceous components mass concentration measurement

The measurement methods of carbonaceous component concentrations can be referred to our previous study (Chen et al., 2023). Briefly, organic carbon (OC) and elemental carbon (EC) are analyzed using a thermal optical transmittance (TOT) carbon analyzer (Sunset Laboratory, Inc., Tigard, OR, USA) with the National Institute for Occupational Safety and Health (NIOSH) protocol. A series of sucrose standard solutions are applied to ensure the status of the instrument and calibrate to obtain the final OC and EC concentrations. Concentrations of primary organic carbon (POC) and secondary organic carbon (SOC) are estimated according to the EC-tracer method and more details can be found in Text S1. A portion of each filter (about 6 cm$^2$) is extracted with ultrapure water (>18.2 MΩ·cm, 25 ℃, Direct-Q, Millipore) by ultra-sonication for 30 min, and then the extract is filtered through a 0.45 μm pore-size PTFE syringe filter (Pall, USA) to remove water-insoluble materials. WSOC concentrations are analyzed using a total organic carbon (TOC) analyzer (Elementar, Germany) and a series of concentration gradient potassium hydrogen phthalate ($C_8H_5O_4K$) standard solutions are used for calibration. Each sample is analyzed in triplicate, and the average of two adjacent values is recorded to minimize the measurement error.

## 2.3 WSOC light absorption spectra measurement and light absorption factor determination

The light-absorbing spectra of WSOC are measured using a spectrophotometer (TIDAS®S 300 UV/VIS 1972 DH, J&M, Germany) coupled with a 1 m path-length liquid waveguide capillary cell (LWCC-3100, World Precision Instruments Inc., USA) over the wavelength range of 250-700 nm with 1 nm scanning interval. Ultrapure water is used as the blank reference during the light absorption spectrum measurement. Light absorption parameters such as light absorption coefficients at 365

nm ($Abs_{365}$), mass absorption efficiency at 365 nm and 405 nm ($MAE_{365}$ and $MAE_{405}$), Ångström exponent over 300-500 nm ($AAE_{300-500}$) are calculated. More details on the calculation methods can be found in Text S2. Furthermore, the direct radiative effect of WSOC is assessed by simple forcing efficiency (SFE) calculated by the measured absorption coefficients in this study following the method described in Text S3. To further analyze the light-absorbing properties of WSOC and compare the discrepancies between different sites, a method based on positive matrix factorization (PMF, EPA PMF 5.0) model combined with light absorption spectra (PMF-LAS) is introduced to analyze the light absorption factor in the main light absorption bands of WSOC (250-500 nm, interval 5 nm). More details about the PMF-LAS method can be found in Text S4.

**2.4 Fluorescent spectra measurement and fluorescence factors determination**

The excitation-emission matrix spectra (EEMs) of WSOC are recorded by a fluorescence spectrometer (Duetta[TM], Horiba Scientific, Japan) in the excitation wavelength (Ex) and emission wavelength (Em) range of 250-600 nm with 5 nm slit width. The intervals of the Ex and Em are 5 nm and 2 nm (4 pixels), respectively. Similarly, ultrapure water is used as the blank reference during the fluorescence spectrum measurement. Fluorescence parameters such as fluorescence index (FI), biological index (BIX) and humidification index (HIX) are calculated according to the method described in our previous study (Chen et al., 2024). More details about the calculation method can be found in Text S5. And fluorophores in WSOC are determined based on the excitation-emission matrix spectra coupled with PARAFAC analysis (EEM-PARAFAC) (Chen et al., 2024).

**2.5 FTIR spectra analysis**

To determine the functional groups in WSOC, an FTIR spectrometer (Nicolet iS50, Thermo Scientific, USA) is employed to obtain the infrared spectrum of WSOC. Briefly, a portion of the filtered extract is freeze-dried and the dried powder is placed on the diamond window of the Attenuated Total Reflection (ATR) platform for measurement. The spectrum is recorded in the average of 128 scans with the wavenumber range of 4000-550 cm[-1] at a resolution of 4 cm[-1]. The background spectrum is collected every 60 minutes. It is worth mentioning that the field blank sample is not measured due to almost no powder in the extract of the blank sample after freeze-drying. The baseline correction and smooth processing of the original spectrum are performed through the OMNIC software (v 9.2), and each functional group is integrated using this software. Relative quantitative of functional groups is achieved by calculating the proportion of each peak area to the total peak area.

**2.6 Relationship and influencing factor analysis**

Multiple linear regression (MLR) model is applied to reveal the relationship between fluorophores and WSOC light absorption and to quantify the contribution of fluorophores to light absorption. To evaluate the contribution of different fluorophores to the light absorption coefficient, ridge regression in MLR model is used to analyze the relationship between the fluorescence intensities of fluorophores and $Abs_{365}$. In the model calculation process, the $Abs_{365}$ is treated as the dependent variable and the maximum fluorescence intensity ($F_{max}$) is introduced as the independent variable, and insignificant ($p < 0.05$) fluorescent

components are excluded from the regression. The calculation is mainly carried out through SPSS software (IBM SPSS Statistics 23). Notably, t-test (two-sample testing under heteroscedasticity, at the 95% confidence level) is conducted to evaluate the significance level of data differences in this study. Besides, the extreme gradient boosting (XGBoost) model is applied to verify the results from MLR (Li et al., 2024). Additionally, the XGBoost model is used to evaluate the influence of conventional gas parameters (e.g., CO, $SO_2$, $O_3$, $NO_2$) on the light absorption of WSOC. More detailed description about the XGBoost method can be found in Text S6.

## 3 Results and discussion

### 3.1 Spatial similarity and heterogeneity of WSOC

### 3.1.1 Concentration levels of WSOC

During the observation period, the mass concentrations of carbonaceous components (i.e., OC, EC and WSOC) increase with the increase of $PM_{2.5}$ concentration (see Figure S3), and exhibit significant spatial variations across the ten sites ($p < 0.05$). As shown in Figure S1 and Table S2, the average concentrations of OC, EC and WSOC observed at the ten sites ranged from 3.31 to 19.6 $\mu g \cdot m^{-3}$, 0.35 to 2.86 $\mu g \cdot m^{-3}$, and 1.97 to 10.6 $\mu g \cdot m^{-3}$, respectively, with the highest mean mass concentrations of OC, EC and WSOC observed in CQ, XA, and HD, respectively, while the lowest values all observed in SH. Overall, the regional average carbonaceous component concentrations show the spatial distribution trends of northwest China > southwest China > north China > east China > regional site ($p < 0.05$). This spatial variability may be attributed to differences in the sources of carbonaceous components and meteorological conditions. For example, the enhanced fuel (e.g., fossil fuels or biomass) combustion for winter heating activities may lead to increased emissions of carbonaceous components at sites in northern China (e.g., HD, XA, TJ and QD) (Wang et al., 2022a; Zhang et al., 2015). The unfavorable meteorological conditions such as low wind speeds may lead to the high concentration levels at sites in southwest China (e.g., CD: 1.16 ± 0.48 $m \cdot s^{-1}$; CQ: 2.27 ± 0.89 $m \cdot s^{-1}$). Moreover, concentrations of carbonaceous components in inland cities (i.e., HD, NJ, XA, CD, CQ) are much higher than those in coastal cities (i.e., TJ, QD, SH) ($p < 0.01$), which is consistent with that reported in previous studies that have shown that air masses from the ocean generally contain lower levels of aerosol content and carbonaceous components (Chen et al., 2023; Diesch et al., 2012; Mo et al., 2022; Zhang et al., 2022b). The 48-h backward air mass trajectory analysis shows that about 1/5 to 1/2 of the air masses that arrive at the coastal cities during the observation period pass through the ocean region, while the inland cities are predominately affected by continental air masses, which may contain a large number of anthropogenic aerosols (see Figure S4). Furthermore, it is worth noting that the regional site TS in NCP has a relatively low mass concentration of carbonaceous components compared to urban sites, which may be due to its high altitude (~1500 m) and low local anthropogenic activities (Jiang et al., 2020). In contrast, the mass concentrations of carbonaceous components at HS (another regional site) in the PRD region are relatively higher compared to TS site. The backward air mass trajectory analysis indicates that more than 80% of the air masses arriving at the HS site originated from the PRD region and are

accompanied by low wind speeds (1.54 m·s$^{-1}$ on average during the sampling period). This suggests that there may be high anthropogenic emissions in the PRD region during the winter sampling period. Over the study period, the OC/EC and WSOC/OC ratios, which can characterize primary sources of carbonaceous aerosols and formation of secondary organic aerosols as well as atmospheric oxidation levels (Ram et al., 2012; Wang et al., 2016), vary from 2.69 to 19.5 (4.58 ± 0.93-10.1 ± 2.69 on average) and 22.7% to 96.1% (53.4 ± 4.94%-73.3 ± 10.0% on average) across the ten sites, indicating non-negligible impacts from both combustion-related emissions and secondary formation on carbonaceous aerosols.

### 3.1.2 Light absorption of WSOC

As shown in Table 1 and Figure S1, the average Abs$_{365}$ (1.12 ± 0.53-13.1 ± 6.95 Mm$^{-1}$) and MAE$_{365}$ (0.56 ± 0.11-1.26 ± 0.34 m$^2$·g$^{-1}$) of WSOC at the ten sites display significant spatial discrepancies ($p < 0.05$), with HD (SH) has the highest (lowest) average Abs$_{365}$ (13.1 ± 6.95 Mm$^{-1}$ (1.12 ± 0.53 Mm$^{-1}$)) and MAE$_{365}$ (1.26 ± 0.34 m$^2$·g$^{-1}$ (0.56 ± 0.11 m$^2$·g$^{-1}$)), respectively. MAE$_{365}$ in SH, CD, TS and HS (0.56 ± 0.11-0.74 ± 0.24 m$^2$·g$^{-1}$ on average) are comparable to those reported in light-polluted areas such as in Guangzhou, Lulang, Waliguan, Urumqi in China and Los Angeles in the USA (0.48-0.81 m$^2$·g$^{-1}$) (Fan et al., 2016; Liu et al., 2018; Soleimanian et al., 2020; Wu et al., 2020; Xu et al., 2020; Zhong et al., 2023). However, they are all lower than those in TJ, HD, QD, NJ, XA and CQ (0.89 ± 0.22-1.26 ± 0.34 m$^2$·g$^{-1}$ on average), which are comparable to those reported in heavy pollution areas such as in Beijing, Xining, Yinchuan, Lanzhou, Taipei in China and Patiala and Mohanpur in India (0.93-1.30 m$^2$·g$^{-1}$) (Cheng et al., 2016; Srinivas et al., 2016; Dey et al., 2021; Zhong et al., 2023; Ting et al., 2022). In this study, the light absorption of WSOC in different region is significantly different ($p < 0.05$), with the regional average Abs$_{365}$ and MAE$_{365}$ displaying as northwest China > southwest China > north China > east China > regional site. Moreover, the average Abs$_{365}$ and MAE$_{365}$ are higher in Northern China (including TJ, HD, QD, and XA, 7.34 ± 5.21 Mm$^{-1}$ and 1.02 ± 0.29 m$^2$·g$^{-1}$, respectively) than in Southern China (including NJ, SH, CD, and CQ, 5.86 ± 3.91 Mm$^{-1}$ and 0.78 ± 0.23 m$^2$ g$^{-1}$, respectively) and regional sites (e.g., TS and HS, 2.91 ± 1.38 Mm$^{-1}$ and 0.72 ± 0.23 m$^2$·g$^{-1}$, respectively) ($p < 0.01$), and higher in inland (8.24 ± 4.75 Mm$^{-1}$ and 0.91 ± 0.27 m$^2$·g$^{-1}$, respectively) than in coastal areas (4.37 ± 3.52 Mm$^{-1}$ and 0.88 ± 0.32 m$^2$·g$^{-1}$, respectively) ($p < 0.01$, see Figure S5), which are consistent with the regional differences in carbonaceous component mass concentrations. This spatial variation may be related to the diversity of WSOC sources and can be affected by the atmospheric processes (including meteorological processes), and is intrinsically related to the chemical composition and structures of WSOC at different sites (Wang et al., 2023; Wang et al., 2024). Previous studies have indicated that the increase in primary emissions such as coal combustion and biomass burning during the winter heating period in Northern China will lead to an enhancement of the WSOC light absorption (Yan et al., 2017; Zhang et al., 2021). The strong correlation between Abs$_{365}$ and POC (r range: 0.59-0.90) or SOC (r range: 0.43-0.97) ($p < 0.01$) (see Figure S6) indicates that light-absorbing components in WSOC are simultaneously affected by both primary emission and secondary formation. The effects of different factors such as chemical composition/structure, and meteorological conditions on the light absorption of WSOC will be discussed in detail in the following sections.

Figure 1 illustrates the log$_{10}$(MAE$_{405}$) and AAE$_{300-500}$ values of WSOC measured in this study and reported in previous studies.

The map in $\log_{10}(MAE_{405})$ vs. $AAE_{300-500}$ space has been proposed by Saleh (2020) to classify BrC types based on light-absorbing properties. Notably, most values measured at the ten sites in this study fall in the regions of weakly-absorbing BrC (W-BrC), which are similar to the values of ambient samples reported in previous studies. However, there are slight differences in the distribution of values between each site. QD has the widest range of $\log_{10}(MAE_{405})$ and $AAE_{300-500}$ values, and is also close to the values of biomass burning samples. Previous studies have indicated that biomass burning (especially those related to residential heating and cooking activities) around the sampling site in QD in winter is a major source contributor to atmospheric particulate matter (Li et al., 2024), and has a significant impact on the light absorption of WSOC (Zhan et al., 2022a; Zhan et al., 2022b). The difference in light absorption of WSOC generated from different fuel (e.g., corn straw, rice straw and wood) combustion may be the reason for the wide range of $\log_{10}(MAE_{405})$ and $AAE_{300-500}$ values in QD (Fan et al., 2016). The values in SH are much closer to very weakly-absorbing BrC (VW-BrC) compared to other sites, which may be related to the influence of marine air mass, as the backward air mass trajectory analysis shows that more than half of the air masses arriving in SH passing through the ocean area during the sampling period (see Figure S4). In contrast, the values in XA are mainly distributed in the moderately-absorbing BrC (M-BrC) region, with a few samples falling in the W-BrC region. This indicates that WSOC in XA has a stronger light-absorbing capacity, which is consistent with its higher $Abs_{365}$ and $MAE_{365}$ values. The average value in XA is much closer to that of coal combustion source samples. Previous studies conducted in XA have shown that the light-absorbing capacity of WSOC in XA region is always at a high level, especially during the winter heating period when anthropogenic emissions such as biomass burning and coal combustion activities increased (Huang et al., 2018; Lei et al., 2019; Yuan et al., 2020). The cause of the high light absorption level of WSOC in XA needs to be further investigated.

Accordingly, the SFE within the wavelength range of 300-400 nm ($SFE_{300-400}$) and 300-700 nm ($SFE_{300-700}$) in different regions are calculated based on the measured MAE values. The $dSFE/d\lambda$ spectra and integrated SFE values are shown in Figure S7. Overall, the integral SFE values exhibit significant spatial variations across the ten sites ($p < 0.01$), and the spatial distribution trend is similar to that of $MAE_{365}$. Among which, HD (SH) has the highest (lowest) $SFE_{300-400}$ ($1.92 \pm 0.51$ W·g$^{-1}$ ($0.88 \pm 0.17$ W·g$^{-1}$)) and $SFE_{300-700}$ values ($4.50 \pm 1.25$ W·g$^{-1}$ ($1.82 \pm 0.38$ W·g$^{-1}$)). The similar variations in SFE and MAE values suggest that the stronger light-absorbing capacity of BrC may lead to an increase in its direct radiative forcing. This is consistent with previous studies, which suggest that more abundant BrC with stronger light-absorbing capacity may result in a remarkable increase in the direct radiative forcing of BrC (Deng et al., 2022; Zhang et al., 2020). It should be noted that $SFE_{300-400}$ accounts for more than 40% ($38.6 \pm 5.04\%$-$48.9 \pm 4.05\%$ on average) of $SFE_{300-700}$ across the ten sites in this study, which is consistent with a previous study (Deng et al., 2022), indicating that the light absorption of WSOC plays a crucial role in the aerosol direct radiative forcing in the UV-Vis range. Notably, there is a significant negative correlation ($p < 0.01$) between SFE value and WSOC/OC at most sites (see Figure S7c and d). The WSOC/OC ratio has been used to infer the degree of secondary aerosol formation or aging of aerosols (Dasari et al., 2019; Ram et al., 2012), therefore, the reduction of SFE value of WSOC may be related to the secondary or aged organic aerosols. Previous studies have indicated that the light absorption capacity of secondary BrC is usually lower than that of primary BrC (Fan et al., 2018; Tang et al., 2020; Zhong and Jang, 2011).

Furthermore, BrC chromophores with aromatic rings and nitro and phenolic groups may undergo photolysis or photochemical oxidation under sunlight, resulting in photo-bleaching and decrease of light absorption capacity and radiation effect of WSOC (Dasari et al., 2019). However, it is noted that this is only a speculation, and further research is needed in the future in conjunction with atmospheric chemical processes and WSOC component analysis.

### 3.1.3 Fluorescence characteristics of WSOC

The fluorescence spectroscopy analysis shows that urban sites (except NJ) have higher humidification index (HIX) values compared to regional sites (see Table 1 and Figure S8), suggesting that the fluorescence components in WSOC at urban sites have higher aromaticity than those at regional sites. The HIX values at urban sites are comparable to those of ambient aerosols and fresh biogenic SOA as well as aged SOA generated from the Maillard reaction reported in previous studies. However, the HIX values measured at the ten sites are overall much lower compared to those of aged biogenic SOA (Lee et al., 2013). This suggests that the contribution of aged biogenic SOA to fluorescence components is relatively small at all site during the wintertime in this study. Moreover, the urban sites in Northern China (such as TJ, QD, XA, HD) exhibit higher biological index (BIX) and fluorescence index (FI) values compared to the urban sites in Southern China (such as CD, CQ, NJ) and regional sites (TS and HS). This suggests that primary combustion emissions such as biomass burning and coal combustion have higher influences on the fluorescent components in Northern China compared to Southern China and regional sites (Tang et al., 2021). It agrees well with different heating modes in north and south China in winter, and is also consistent with the pollution sources at the regional sites.

Different fluorophores in WSOC are identified with the EEM-PARAFAC method. Five fluorophores (C1-C5) are resolved in TJ, four fluorophores (C1-C4) are resolved in QD, SH and HS, and three fluorophores (C1-C3) are resolved at other sites, respectively (see Figure S9). Therein, C1 at urban sites (TJ, HD, QD, NJ, SH, XA, CD, CQ) has a primary (secondary) Ex peak at around 250 nm (300 nm) with Em peak at around 395 nm, while Ex peak of C1 at regional sites (TS and HS) is at around 250 nm and 300 nm or 315 nm with Em peak at around 413 nm. Such fluorophores can be classified as less-oxygenated species (LO-HULIS) (Chen et al., 2016b). Moreover, the C4 resolved in TJ, QD, SH and HS are also considered as LO-HULIS for its main peak is distributed in the LO-HULIS region (Chen et al., 2020). Previous studies have reported that LO-HULIS is abundant in biomass burning aerosols, especially in winter (Jiang et al., 2022a). Additionally, industrial sources and other combustion related sources also make important contributions to LO-HULIS (Chen et al., 2020). C2 and C5 (only found in TJ) are with longer emission wavelengths (Em > 420 nm) and contain multiple Ex peaks (e.g., 250, 295, 305, 340, 345, 350, 355, 360, 370, 375 nm), and these fluorophores are regarded as highly-oxygenated species (HO-HULIS) (Chen et al., 2020). The multiple Ex peaks and longer Em peaks of HO-HULIS may be associated with aromatic conjugated structures and may contain heteroatoms (Chen et al., 2016a). HO-HULIS generally has a strong correlation with anthropogenic secondary formation and combustion emission (e.g., biomass burning and coal combustion) aerosols (Jiang et al., 2022a; Jiang et al., 2022b; Li et al., 2023b). Differently, C3 is mainly distributed in areas with lower Ex (260-275 nm) and Em (285-336 nm) wavelengths. Recent studies have shown that these fluorophores are compatible with the fluorescent peaks of amino acids (e.g., tyrosine, tryptophan)

and non-N aromatic species (non-Nas, e.g., aromatic acids, phenolic compounds and their derivatives) (Cao et al., 2023; Chen et al., 2020). As concentrations of atmospheric amino acids are usually negligible, and the corresponding contribution to fluorophores is insignificant when compared to non-Nas (Cao et al., 2023; Chen et al., 2020). Therefore, C3 is defined as non-Nas in this study, which is more likely derived from fossil fuel combustion (Li et al., 2023b; Tang et al., 2020).

Figure 2 illustrates the average fluorescence volume (FV) and relative contributions of different fluorophores at different sites. Clearly, the average FV of all fluorophores varies in the range of $2.50\times10^2$-$9.76\times10^3$ RU-nm$^2$, showing great spatial variability. Overall, the LO-HULIS fluorophore has the highest FV ($1.49\times10^3 \pm 4.74\times10^2$-$9.76\times10^3 \pm 5.82\times10^3$ RU-nm$^2$ on average) and accounts for the largest proportion ($42.2 \pm 5.59\%$-$60.3 \pm 2.11\%$ on average) in the total fluorophores at most sites (except for TS, NJ and CD), demonstrating the widespread existence of combustion-related sources (especially biomass burning) and their important influence on the fluorophore. In contrast, HO-HULIS fluorophore accounts for the lowest proportion ($12.5 \pm 1.32\%$-$23.8 \pm 2.43\%$ on average, except for TJ and TS). This suggests that the effect of secondary anthropogenic sources at different location may be relatively small during the winter study period. The relatively high FV values of LO-HULIS and non-Nas in northern cities ($1.62\times10^6$ RU-nm$^2$ in total) compared to southern cities ($4.37\times10^5$ RU-nm$^2$ in total) and regionals sites ($1.76\times10^5$ RU-nm$^2$ in total) further indicate the impact of increased primary emissions during the heating season, especially in northern China (Cao et al., 2023; Li et al., 2023b). This is consistent with a previous study by Cao et al. (2024a), which showed that the FV of fluorophores in BrC during the winter heating season was significantly higher than that during the non-heating season.

### 3.1.4 Functional group structural characteristics of WSOC

The structure of WSOC is further investigated by FTIR spectroscopy. The spectra of WSOC are generally similar at different sites (see Figure 3). The FTIR spectra at each site mainly include six or seven absorption bands at 2635-3600 cm$^{-1}$, 1540-1820 cm$^{-1}$, 1220-1510 cm$^{-1}$, 977-1220 cm$^{-1}$, 860-960 cm$^{-1}$, 806-844 cm$^{-1}$ and 590-727 cm$^{-1}$. The peak in the widest band at 2635-3600 cm$^{-1}$ can be attributable to the intramolecular and intermolecular O-H stretching vibrations of alcohols, phenols and carboxylic acid (Fan et al., 2023; Huang et al., 2022; Wang et al., 2021). The peak in the band of 1540-1820 cm$^{-1}$ is recognized as C=O stretching vibrations of carboxylic acids, ketones, aldehydes and esters (Fan et al., 2023; Wang et al., 2021; Yang et al., 2024). The sharp peak in the range of 1220-1510 cm$^{-1}$ is attributed to C=C stretching vibrations of aromatic rings (Wang et al., 2021). The strongest peak occurs in the band of 977-1220 cm$^{-1}$ can be assigned to C-O stretching vibrations of phenols, esters and ethers (Fan et al., 2023; Wang et al., 2021). The peak in the range of 860-960 cm$^{-1}$ is only observed at some sites (i.e., TJ, HD, QD, CD and TS), which can be ascribed to C=C-H in alkenes (Yu et al., 2018). And the sharpest peak in the band of 806-844 cm$^{-1}$ corresponds to R-ONO$_2$ stretching of organic nitrate (Huang et al., 2022), and multiple small peaks in the band at 590-727 cm$^{-1}$ represent C-H bending vibrations of aromatic rings (Fan et al., 2023; Wang et al., 2021).

Quantitative analysis of functional groups is conducted by integrating peak area and the proportion of different functional groups at each site is presented in ring charts in Figure 3. Totally, the proportion contribution shows that O-H is the most

abundant at all sites (39.2 ± 4.57%-48.4 ± 3.12% on average), followed by C=C (16.6 ± 4.42%-30.9 ± 3.23% on average) and C-O (14.0 ± 4.48%-27.8 ± 4.65% on average). The proportion of C=C is generally higher at sites in Southern China than in Northern China, while C-O is the opposite ($p < 0.05$). O-H and C=C are negatively correlated with $E_2/E_3$ (the ratio of the light absorption of WSOC at 250 nm and 365 nm, see Figure S10a and b). This indicates that these two functional groups may originate from aromatic compounds with higher molecular weight and higher degree of aromaticity, as lower $E_2/E_3$ values are related to higher aromaticity and molecular weight (Peuravuori and Pihlaja, 1997). To a certain extent, this can explain the strong light absorption capacity of WSOC in Northern China in winter. In contrast, the proportion of C-O and C=C-H (only detected at some sites) is positively correlated with $E_2/E_3$ (see Figure S10c and d), indicating that these two functional groups are mainly related to ester or ether and aliphatic hydrocarbon compounds with smaller molecular weights. The proportion of these two functional groups is significantly higher at the regional site (TS) in Northern China than at other sites ($p < 0.05$). In contrast, C=O makes a greater contribution at the regional site (HS) in Southern China.

### 3.2 Identification of light-absorbing substances based on light absorption spectra

In order to further explore the light absorption characteristics of WSOC and their diversity at different sites, the absorption spectra (250-700 nm) of WSOC are analyzed. Clearly, the light absorption coefficients ($Abs_\lambda$) of WSOC at each site exhibit strong wavelength dependence, especially over the UV-visible range (250-500 nm) where there is a significant light absorption signal. According to the UV-visible absorption spectra (see Figure 4), the WSOC absorption spectra measured at the ten sites can be classified into two categories. That is, one type with light absorption continues to decline from 250 nm to 700 nm, with a peak around 250 nm (namely unimodal type), and the other type with two significant absorption peaks at 265 nm and 300 nm (namely bimodal type). Interestingly, these two types of spectra happen to correspond to the sites in East China (unimodal: QD, NJ, SH and TS) and those outside East China (bimodal: TJ, HD, XA, CD, CQ and HS), respectively. The differences in the spectral types may be related to differences in the light-absorbing species present in WSOC. Therefore, the PMF-LAS method is further used to analyze the potential categories of the light-absorbing substances. Based on this method, the unimodal and bimodal spectra measured in East China sites and outside East China sites are separately put into the PMF model, and finally three different light absorption factors (namely uni-Fac1, uni-Fac2, uni-Fac3; and bi-Fac1, bi-Fac2, bi-Fac3) are resolved, respectively (see Figure 5 and Figure S11).

By comparing with the light absorption spectra of light-absorbing species in previous studies, the spectra resolved in this study are found to be similar to those of most aromatic or nitrogen-containing heterocyclic compounds. For example, bi-Fac1 exhibits a clear absorption peak at 350 nm and possibly stronger absorption peaks below 250 nm, which is consistent with the spectra of nitro-aromatic compounds such as 4-nitrocatechol (Huang et al., 2021; Lin et al., 2018; Yang et al., 2023). The absorbance of uni-Fac1, uni-Fac2 and bi-Fac2 decreases sharply with increasing wavelength. The uni-Fac2 and bi-Fac2 exhibit an absorption peak at 260 nm, while uni-Fac1 shows no distinct peaks. These factors are similar to the absorption spectra of nitro-aromatic compounds and nitrogen-free aromatic compounds, respectively (Cao et al., 2023; Jiang et al., 2022a). Additionally, uni-Fac3 and bi-Fac3 have a main absorption peak at around 310 nm and 305 nm, which match the absorption peak of most

nitrogen-free aromatic compounds such as vanillin and a few nitro-aromatic compounds or nitrogen-containing heterocyclic compounds (Huang et al., 2021; Li et al., 2020b; Lin et al., 2018; Yang et al., 2023). Taken together, the above spectral analysis suggests that the important light-absorbing components in WSOC may be mainly aromatic compounds or nitrogen-containing compounds. However, it is worth noting that this judgment is based on substances with known absorption spectra, and further studies on more kinds of absorption spectra are needed in the future.

The contributions by different light absorption factors vary significantly at different wavelengths (see Figure S12). For sites locating in East China, the light absorption contribution by uni-Fac1 gradually decreases with the increase of wavelength with the largest contribution at 250 nm. In contrast, the absorption contribution of uni-Fac3 increases significantly over 250-325 nm and then slowly over 325-500 nm. The absorption contribution of uni-Fac2 decreases first and then increases, with the minimum contribution appearing around 340 nm. For sites in outside East China, the contribution of bi-Fac1 is relatively stable in the 250-300 nm range, but increases significantly with wavelength at the wavelengths above 300 nm. In contrast, the proportional contribution of bi-Fac2 is similar to that of uni-Fac1, which monotonically decreases with increasing wavelength throughout the entire spectral range (250-500 nm). The proportional contribution of bi-Fac3 significantly increases in the range of 250-320 nm, and then remains stable or slightly decreases thereafter. Overall, uni-Fac1 is the main absorption factor at sites in East China, while bi-Fac3 is the major absorption factor at sites outside East China, contributing to $38.9 \pm 10.4\%$-$53.5 \pm 13.5\%$ and $39.5 \pm 22.7\%$-$51.1 \pm 20.0\%$ (on average) of the total light absorption, respectively. This suggests that nitro-aromatic or nitrogen-free aromatic compounds with strong wavelength dependence are the main light-absorbing species in WSOC at the ten sites, highlighting the importance of aromatic structure to WSOC light absorption. However, it is worth noting that this is only preliminary knowledge based on substances with known absorption spectra. The three types of light-absorbing factors may contain different aromatic and other light-absorbing substances, which may have different relative contributions, and therefore present significantly different light absorption spectra. In the future, it is necessary to combine mass spectrometry techniques to explore the composition of light-absorbing substances in different classes at the molecular level.

The $E_2/E_3$ ratio calculated based on the light absorption spectra also shows that aromatic compounds are important light absorbers. As shown in Figure 6, $E_2/E_3$ values in north China, northwest China, and southwest China are relatively low, especially in XA, while the $E_2/E_3$ value at TS site is the highest. A very strong negative correlation between the $Abs_{365}$ (*r* range: -0.96 to -0.32, *p < 0.05*), $MAE_{365}$ (*r* range: -0.83 to -0.32, *p < 0.05*) and $E_2/E_3$ values are observed. This further suggests that the strong light-absorbing ability of WSOC may be associated with more aromatic structures. A strong negative correlation (*p < 0.05*) is also found between $E_2/E_3$ and fluorophores (especially HULIS fluorophore) (see Figure S13), suggesting that the fluorophores also contain aromatic structures, and there may be a certain correlation between fluorophores and light-absorbing components, which will be discussed in the subsequent section.

### 3.3 Influencing factors of optical properties of WSOC

#### 3.3.1 Variations of optical properties of WSOC with concentrations of $PM_{2.5}$ and gaseous pollutants

To investigate the influence of air quality levels on the light absorption properties of WSOC, the sampling days are classified into five pollution levels including clean (0-35 µg·m$^{-3}$), relatively clean (35-75 µg·m$^{-3}$), slightly polluted (75-115 µg·m$^{-3}$), moderately polluted (115-150 µg·m$^{-3}$), and heavily polluted (> 150 µg·m$^{-3}$) according to the national ambient air quality daily Grade-II standard threshold values and ambient air quality indices. As shown in Figure 7a and b, $Abs_{365}$ and $MAE_{365}$ of WSOC both increase with the increase of pollution levels, in which $Abs_{365}$ changes significantly ($p < 0.01$) while $MAE_{365}$ changes relatively gently. The enhancement of WSOC light absorption under high pollution conditions may be related to the increase of WSOC concentration, light absorption capacity and light-absorbing species. Previous studies have reported that the mass fractions of oxidized organic aerosols increase significantly with the increase of $PM_{2.5}$ mass concentration, and the oxidized organic aerosols contain a large number of light-absorbing species such as nitroaromatics compounds (You et al., 2024). In this study, the relative abundances of O-H, C=C and $R-ONO_2$ functional groups, which are related to aromatic compounds and have a good positive correlation with the light absorption of WSOC, increase with $PM_{2.5}$ mass concentration (see Figure S14, and discussion in the next section). Additionally, the accumulation of anthropogenic emissions (especially those sources with strong light-absorbing BrC such as biomass burning and coal combustion sources) at high pollution levels will lead to an increase in BrC chromophore types and overall light absorption capacity (Li et al., 2020a; Tang et al., 2020; Wei et al., 2020). The fluorescence volumes normalized (NFV) by WSOC concentration of different fluorophores exhibit different variation trends with $PM_{2.5}$ mass concentrations (see Figure 7c-e). Overall, the total NFV value of HULIS increases with $PM_{2.5}$ concentrations, with the increase of HO-HULIS being more monotonous and significant ($p < 0.01$, based on Spearman's rank correlation test) while the increase of NFV of LO-HULIS being less significant ($p > 0.05$). In contrast, the NFV of non-Nas fluorophore decreases with the increase of $PM_{2.5}$ concentrations. This suggests that HO-HULIS is the dominant fluorophore under contaminated conditions (see Figure S14). The different degrees of increase in HO-HULIS and LO-HULIS highlight the contributions of combustion related sources and secondary sources and the increase of aerosol oxidation under high pollution levels. This also implies an increase in chromophores with aromatic or heterocyclic structures under pollution conditions, which is consistent with the indication of functional groups.

The relationships between conventional air pollutants (CO, $NO_2$, $SO_2$, $O_3$) and $Abs_{365}$ are also analyzed. As shown in Figure S15, a strong positive correlation ($r$ range: 0.40-0.93, $p < 0.05$) between $Abs_{365}$ and CO, $NO_2$, $SO_2$ is observed in TJ, QD, SH, XA, CD, CQ and TS. As such air pollutants usually originate from fossil fuel combustion (e.g., coal combustion and vehicle emissions) or biomass burning (Adam et al., 2021), which suggests that the light absorption of WSOC in these regions could be influenced by primary combustion. Among these air pollutants, CO and $SO_2$ have the greatest impact on $Abs_{365}$ of WSOC (Figure S15c). A good negative correlation ($r$ range: -0.66 to -0.40, $p < 0.05$) between $Abs_{365}$ and $O_3$ is observed at TJ, QD, SH and XA. $O_3$ is typically considered a type of secondary air pollutant that can be generated through various pathways such as photochemical reactions (Adam et al., 2021). That means, the light absorption of WSOC at these sites may be also affected by atmospheric chemical processes (e.g., photo-bleaching) in addition to the primary combustion.

### 3.3.2 Variations of light absorption properties of WSOC with relative humidity (RH)

Figure 7f shows that $MAE_{365}$ significantly decreases with increase of RH when RH is lower than 60%, while it exhibits no significant changes when RH > 60%. Previous studies have indicated that the mixed particles containing 4-nitrophenol and ammonium sulfate will undergo phase separation, and the separated core-shell particles may enhance light absorption through the lensing effect at low RH (Liu et al., 2023; Price et al., 2022). Moreover, compared to dry conditions, the loss of WSOC light absorption is faster at high RH. It has been reported that non-phenolic aromatic carbonyls may undergo aqueous-phase photo-oxidation to produce $H_2O_2$, which can be further decomposed to produce OH radicals at high RH, and OH radicals will bleach the BrC chromophore (Anastasio et al., 1997; Faust, 1994; Zellner et al., 1990; Zhong and Jang, 2014). Therefore, the decreasing $MAE_{365}$ with increased RH may be related to phase changes, photo-bleaching processes as well as the changes in BrC chromophores.

## 3.4 Relationships among fluorescent chromophore, functional groups and light absorption of WSOC

To better understand the relationships among light absorption, fluorescence and functional group structure of WSOC, several correlation analysis methods are used in this study. Figure S13 shows that $Abs_{365}$ strongly correlated with the TFV of fluorophores at all sites ($p < 0.01$), indicating that the fluorescent components may contribute to WSOC light absorption to some extent. Then, MLR model is further applied to evaluate the relationship between the $F_{max}$ of various fluorescent components and $Abs_{365}$, and a good linear fitting relationship is found between measured $Abs_{365}$ and modeled $Abs_{365}$ (see Figure 8). The adjusted $R^2$ values of the MLR model range from 0.62 to 0.93, implying that the fluorophores identified at the ten sites can explain 61.8%-93.0% of the measured WSOC light absorption. Furthermore, the contribution of various fluorophores to $Abs_{365}$ is quantified by calculating the standardized regression coefficients (*Beta*) in the MLR model (see the pie chart in Figure 8). The results show that HULIS fluorophores have the greatest contribution to $Abs_{365}$, with higher contributions by HO-HULIS (28.5%-49.7%) than LO-HULIS (15.2%-59.6%), and followed by non-Nas (0.31%-25.8%), although HO-HULIS accounts for the lowest proportion of the total fluorophores. The high contribution of HO-HULIS to $Abs_{365}$ may be related to its complex source and composition, such as the presence of unsaturated and high molecular weight aromatic structures in HO-HULIS. Previous studies have shown that the light-absorbing ability of BrC is positively correlated with the unsaturation and molecular weight of chromophores (Chen and Bond, 2010; Tang et al., 2020). It should be noted that HO-HULIS is not resolved at TS site, and the contribution of LO-HULIS to $Abs_{365}$ at TS site is much higher than that of non-Nas (59.6% > 2.2%). To better understand the importance of fluorophores to WSOC light absorption, the fluorescence intensity of each fluorophore at all sites is summarized and input into the XGBoost-SHAP model. The absolute value of the average SHAP score is used to evaluate the importance of different fluorophores on $Abs_{365}$, with the larger the value, the greater its importance. Figure S16 shows that HO-HULIS is of the highest importance, much higher than non-Nas and LO-HULIS. This further demonstrates the important contribution of fluorescent components to light absorption of WSOC, especially by HO-HULIS.

Figure 9 illustrates the relationships of light absorption parameters and fluorescence chromophores with functional groups. In general, fluorescence chromophores (especially the HULIS classes) and the $Abs_{365}$ exhibit similar correlations with functional

groups at all sites, which agrees well with the important contribution of fluorescence chromophores to the light absorption of WSOC as aforementioned. At most sites (except XA and CQ), O-H, C=C and R-ONO$_2$ functional groups show the strongest positive correlations with Abs$_{365}$ and fluorescent chromophores, implying that aromatic compounds (especially at polluted sites such as HD, NJ, QD, and CD) and organic nitrates (especially at less polluted sites such as SH, TS and HS) have important impacts on the optical properties of WSOC. However, ester or ether and aliphatic hydrocarbon functional groups (e.g., C-O and C=C-H) have relatively smaller positive impacts on the optical properties of WSOC, even exhibiting negative correlations with light absorption parameters and fluorescence chromophores at some sites. In contrast, the relationships of functional groups with MAE$_{365}$ are slightly different from that with Abs$_{365}$. Overall, C=C, O-H and R-ONO$_2$ exhibit the strongest correlations with MAE$_{365}$ at most sites (e.g., TJ, QD, SH, TS, and HS). C=O and C-O, which may be related to carboxylic acids, phenols and esters, show positive correlations with MAE$_{365}$ at some sites (e.g., HD, NJ, XA and CQ). The positive correlation of C=O and C-H functional groups with both Abs$_{365}$ and MAE$_{365}$ and the strong positive correlation of C-O group with MAE$_{365}$ in XA could explain the particularity of light absorption characteristics and the ability of WSOC in XA to a certain extent.

## 4 Conclusions

Based on the same measurement methods and data processing processes, light absorption, fluorescence and FTIR spectra analysis are combined to investigate the optical properties and functional group characteristics of WSOC at ten sites in different regions of China. The spatial variations at various sites and the relationships between absorbance, fluorescence, and functional groups of WSOC are revealed. Mass concentrations of carbonaceous components (OC, EC and WSOC) and light absorption of WSOC exhibit a significant spatial variation at the ten sites, which generally manifested as northwest China > southwest China > north China > east China > regional site, with higher values in Northern China than Southern China and regional sites, and higher in inland areas than coastal areas.

The aromatic and large molecular weight structure has a significant impact on the light-absorbing ability of WSOC according to the E$_2$/E$_3$ ratio, PMF-LAS resolved light absorption factors and functional groups. PMF-LAS method determines  three (uni-Fac1~uni-Fac3, bi-Fac1~bi-Fac3) light absorption factors for unimodal light-absorbing spectra at sites in East China and bimodal light-absorbing spectra at sites outside East China, respectively. These factors are mainly aromatic compounds by comparing the spectra with existing BrC species. Furthermore, FTIR based function groups show that aromatic O-H (39.2%-48.5%) and C=C (16.6%-30.9%), as well as aliphatic C-O (14.0%-27.8%) are three most abundant functional groups in WSOC. The strong positive correlation between aromatic O-H and C=C with Abs$_{365}$ and MAE$_{365}$ suggests that aromatic components (especially phenolic compounds) play an important role in the light absorption of WSOC. Three types of fluorophores in WSOC are resolved by EEM-PARAFAC analysis, with LO-HULIS as the most abundant fluorophore, followed by non-Nas and HO-HULIS. The positive correlation between E$_2$/E$_3$ or aromatic functional groups (O-H and C=C) and these fluorophores indicates that aromatic structures also have an important impact on the fluorescent components. This also indicates that there

is an undeniable connection between the light absorption and fluorescence of WSOC. Quantitatively, MLR results show that the identified fluorophores (especially HULIS such as LO-HULIS and HO-HULIS) contribute significantly to the $Abs_{365}$ of WSOC (with proportion contribution of 61.8%-93.0%), with the largest contribution by HO-HULIS (28.5%-49.7%).

Analysis of the relationships between WSOC light absorption and gaseous precursors and meteorological conditions show that primary combustion sources have significant impacts on WSOC light absorption at most sites (such as TJ, QD, SH, XA, CD, CQ and TS), and atmospheric chemical process (such as photobleaching) exhibit more effects on WSOC light absorption at TJ, QD, SH and XA sites. Moreover, the multi-site observation dataset shows that $MAE_{365}$ of WSOC generally increases with pollution levels and decreases with increasing RH within the range of 20%-60% and keeps stable when RH > 60%. Taken

together, this study promotes our knowledge of optical properties and structural characteristics of WSOC in different regions of China and deepened the understanding of the contribution of WSOC fluorescence to its light absorption. In the future, it is necessary to build a quantitative parametric relationship between light absorption, composition and structure of WSOC, especially with the evolution of atmospheric processes. Additionally, it is important to note that this study only focused on WSOC. Since the WISOC may have stronger light absorption capacity, further research on light absorption, composition and

structure of WISOC (especially in northern China in winter) and the correlation between them are also needed in the future.

**Data availability.** Data presented and used throughout this study can be accessed from the following data repository: https://doi.org/10.5281/zenodo.14193028 (Chen, 2024).

**Supplement.** The supplement includes six texts (Texts S1-S6), two tables (Tables S1-S2) and sixteen figures (Figure S1-S16) related to the paper.

**Author contributions.** CY designed the research and supported funding the observation. XW, QC, MX, JL, FW, SF, QY, HN, MZ, YW and LX had an active role in supporting the sampling work. HC carried out the sample pretreatment and instrumental analysis under the guidance of CY, LH and LD. HC processed data, plotted the figures, and wrote the manuscript. YY provided assistance in data processing. CY edited the manuscript. All authors contributed to the discussions of the results and the refinement of the manuscript.

**Competing interests.** The authors declare no conflicts of interest.

**Financial support.** This work was supported by the Natural Science Foundation of China (NO. 4225110), Natural Science Foundation of Shandong Province (ZR2021MD033), Natural Science Foundation of Jiangsu Province (BK20220275), the Excellent Young Scholar (Overseas) project of Shandong Province (2022HWYQ-049), Open fund by Jiangsu Key Laboratory

of Atmospheric Environment Monitoring and Pollution Control (KHK2106), and Taishan Scholars Program of Shandong Province (NO. tsqn201909018) and Qilu Youth Talent of Shandong University.

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

## Tables and Figures

**Table 1.** Light absorption and fluorescence parameters of WSOC measured in this study.

| Sites | Regions | $Abs_{365}$ (Mm$^{-1}$) Avg ± SD | $MAE_{365}$ (m$^2 \cdot$g$^{-1}$) Avg ± SD | $AAE_{300-500}$ Avg ± SD | FI Avg ± SD | BIX Avg ± SD | HIX Avg ± SD |
|---|---|---|---|---|---|---|---|
| Tianjin (TJ) | North China | 5.57 ± 3.83 | 0.89 ± 0.22 | 6.58 ± 0.74 | 1.48 ± 0.06 | 0.97 ± 0.06 | 2.91 ± 0.37 |
| Handan (HD) | | 13.1 ± 6.95 | 1.26 ± 0.34 | 5.96 ± 0.36 | 1.55 ± 0.08 | 1.00 ± 0.11 | 1.07 ± 0.43 |
| Qingdao (QD) | East China | 4.80 ± 3.09 | 1.03 ± 0.34 | 7.04 ± 0.89 | 1.58 ± 0.09 | 1.08 ± 0.11 | 1.69 ± 0.32 |
| Nanjing (NJ) | | 6.26 ± 3.26 | 0.89 ± 0.25 | 6.66 ± 0.35 | 1.49 ± 0.17 | 0.82 ± 0.12 | 0.56 ± 0.20 |
| Shanghai (SH) | | 1.12 ± 0.53 | 0.56 ± 0.11 | 7.02 ± 0.54 | 1.57 ± 0.09 | 1.02 ± 0.08 | 1.98 ± 0.26 |
| Xi'an (XA) | Northwest China | 10.6 ± 4.42 | 1.04 ± 0.11 | 4.83 ± 1.01 | 1.58 ± 0.08 | 0.96 ± 0.09 | 1.53 ± 0.33 |
| Chengdu (CD) | Southwest China | 5.99 ± 2.61 | 0.65 ± 0.11 | 6.24 ± 0.47 | 1.54 ± 0.10 | 0.80 ± 0.07 | 1.18 ± 0.25 |
| Chongqing (CQ) | | 10.6 ± 4.10 | 0.98 ± 0.09 | 5.78 ± 0.39 | 1.51 ± 0.04 | 0.84 ± 0.04 | 1.37 ± 0.24 |
| Mt. Tai (TS) | Regional site | 2.66 ± 1.22 | 0.74 ± 0.24 | 7.11 ± 0.46 | 1.31 ± 0.17 | 0.75 ± 0.15 | 0.26 ± 0.11 |
| Heshan (HS) | | 3.76 ± 1.55 | 0.64 ± 0.12 | 6.28 ± 0.30 | 1.49 ± 0.03 | 0.77 ± 0.06 | 0.91 ± 0.15 |

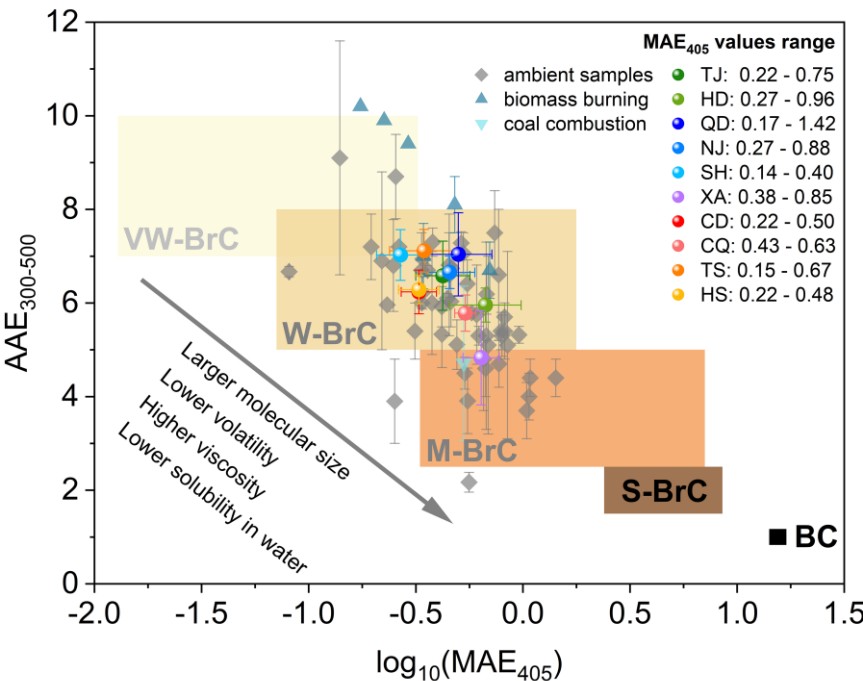

**Figure 1.** Graphical representation of optical-based BrC classes in $\log_{10}(MAE_{405})$-AAE space. The shaded regions represent very weakly light-absorbing BrC (VW-BrC), weakly light-absorbing BrC (W-BrC), moderately light-absorbing BrC (M-BrC), strongly light-absorbing BrC (S-BrC), and absorbing BC, respectively. Date source: Bosch et al., 2014; Chen et al., 2018; Cheng et al., 2011; Cheng et al., 2016; Choudhary et al., 2021; Deng et al., 2022; Dey et al., 2021; Fan et al., 2016; Fang et al., 2023; Hecobian et al., 2010; Huang et al., 2018; Kirillova et al., 2014; Kirillova et al., 2016; Li et al., 2023a; Li et al., 2019; Li et al., 2023c; Liu et al., 2018; Liu et al., 2019; Soleimanian et al., 2020; Srinivas and Sarin, 2013; Srinivas and Sarin, 2014; Srinivas et al., 2016; Tang et al., 2021; Ting et al., 2022; Wu et al., 2020; Xie et al., 2020; Xu et al., 2020; Yan et al., 2015; Yan et al., 2017; Yang et al., 2020; Yuan et al., 2020; Yue et al., 2022; Zhao et al., 2022; Zhong et al., 2023; Zhu et al., 2018

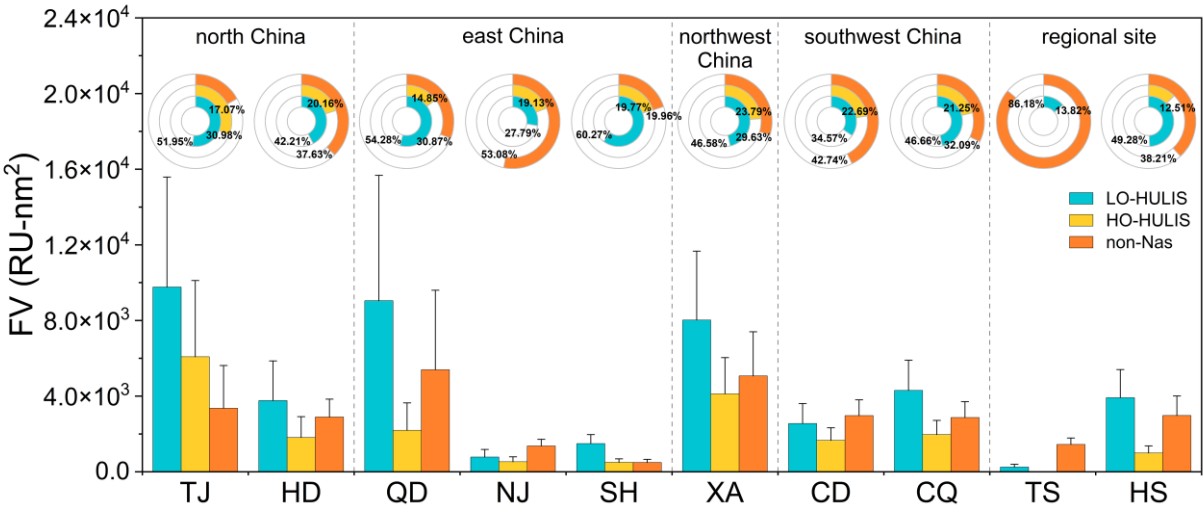

**Figure 2.** The absolute fluorescence volumes (bar chart) and fractional contributions (pie chart) of each fluorophore at different observation sites. Note: Blue denotes LO-HULIS, yellow denotes HO-HULIS, and orange denotes non-Nas.

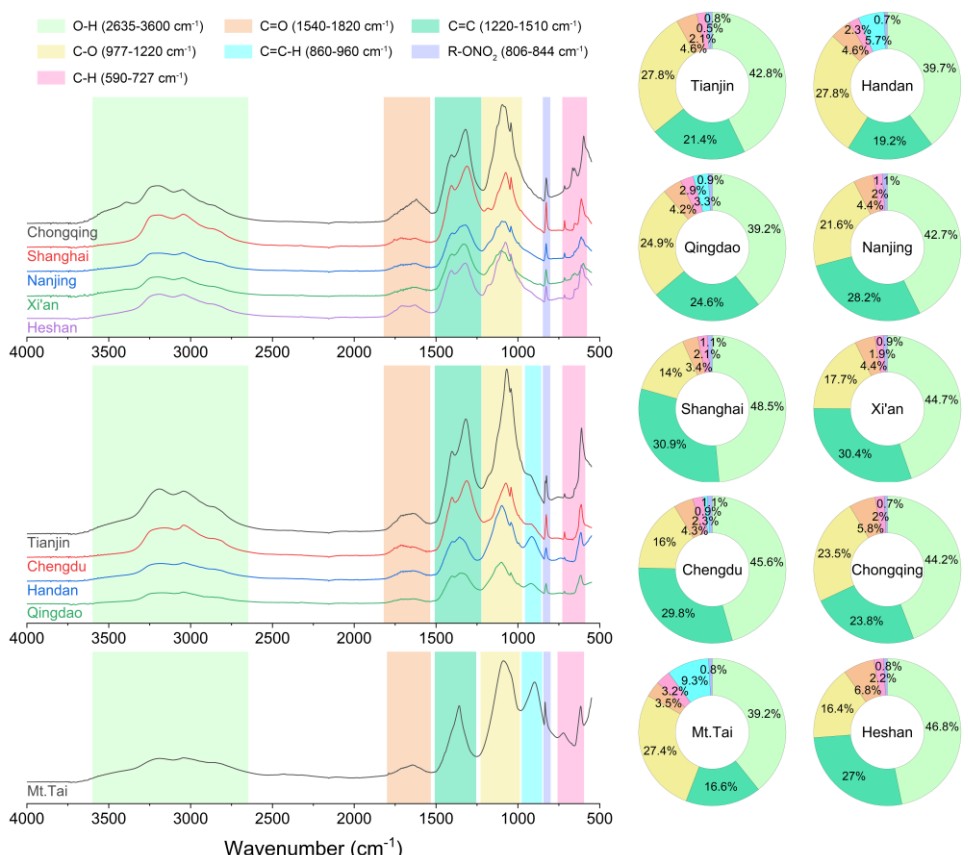

**Figure 3.** FTIR spectra of WSOC and relative proportions of different functional groups measured (ring charts) at ten sites in this study.

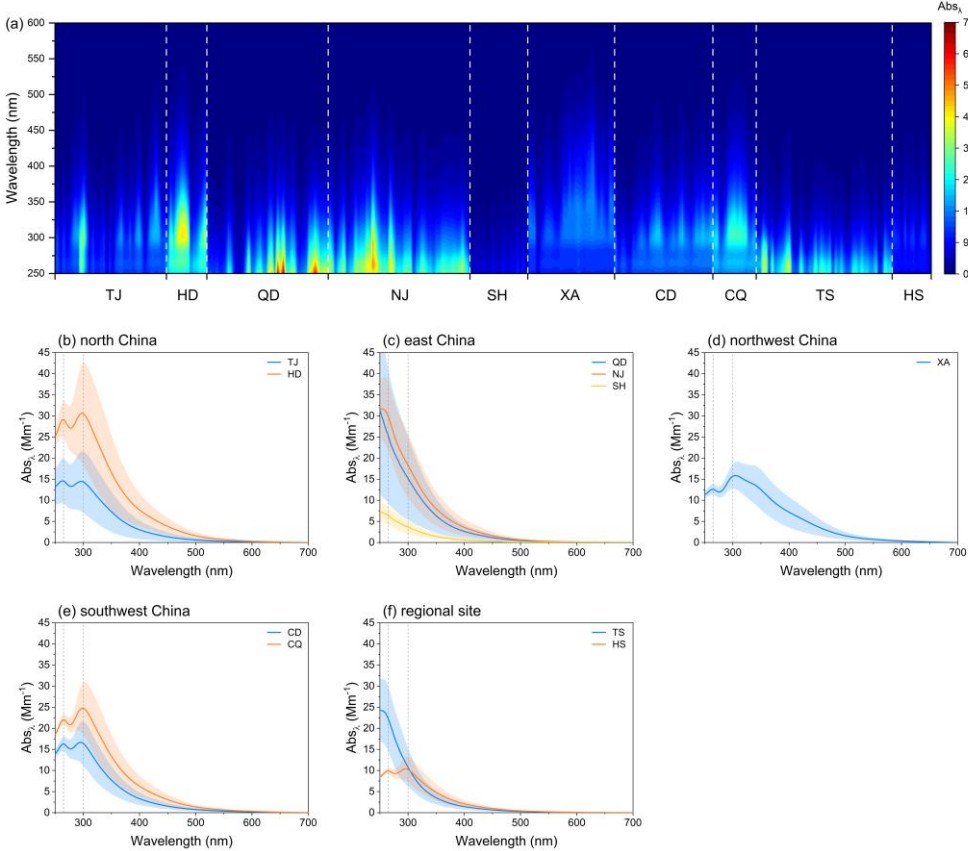

**Figure 4.** (a) Light absorption coefficient spectrum of WSOC at each site and the average light absorption spectrum in (b) north China, (c) east China, (d) northwest China, (e) southwest China, and (f) regional sites. Note: The color bar represents the magnitude of the $Abs_\lambda$.

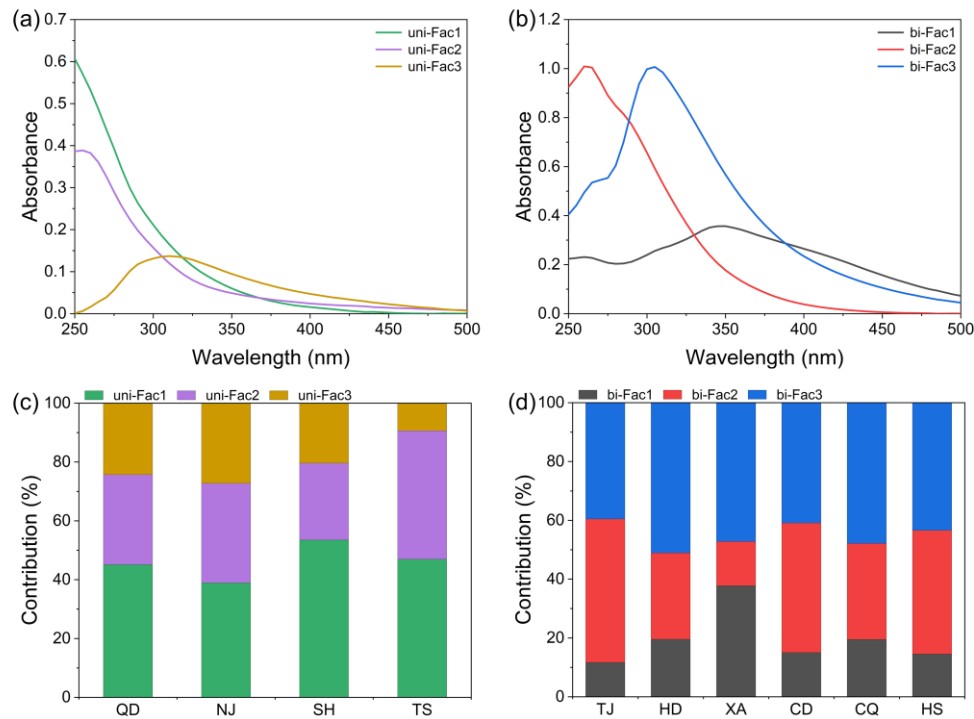

**Figure 5.** The average light absorption spectra of the absorption factors resolved by PMF model at (a) East China sites (unimodal absorption spectral type) and (b) outside East China sites (bimodal absorption spectral type), as well as the average contribution by each factor calculated according to the integral absorbance from 250-500 nm at both types of sites (panel c and d).

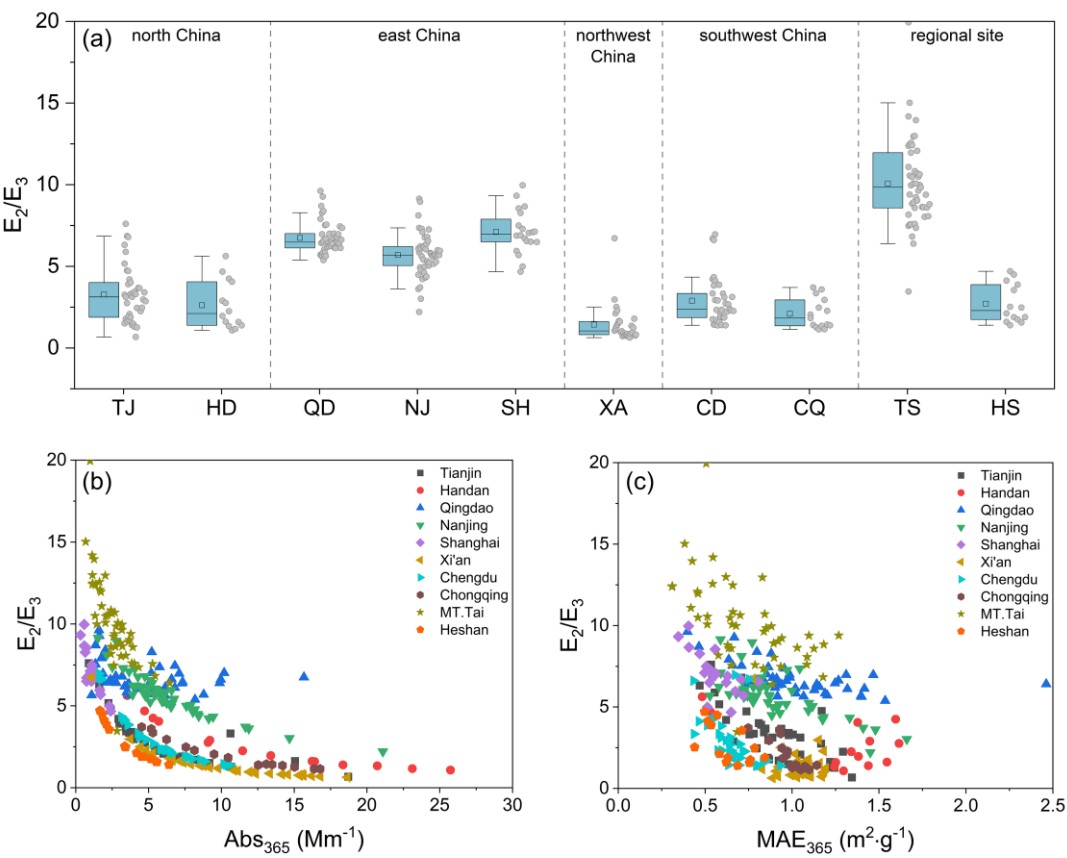

**Figure 6.** Box-plot of $E_2/E_3$ at ten sites (panel a) and scatter plots of relationships between $Abs_{365}$ ($MAE_{365}$) and $E_2/E_3$ (panel b and c).

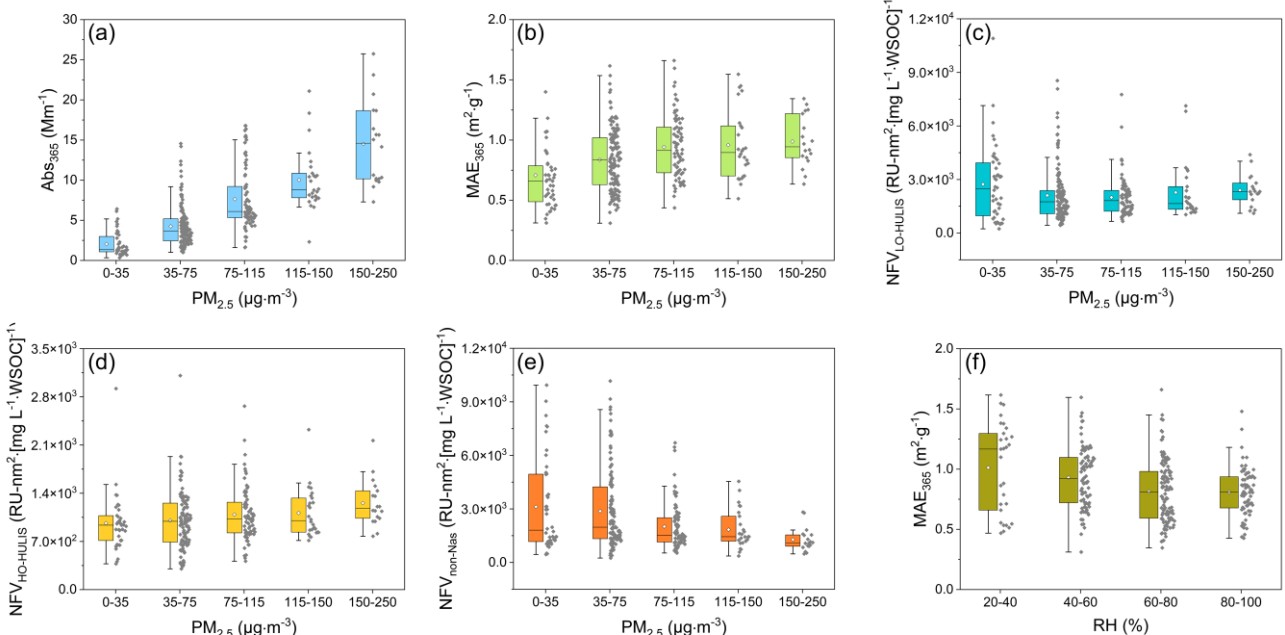

**Figure 7.** Variations of Abs$_{365}$ and MAE$_{365}$ (panel a and b), and NFV of different fluorophores (panel c, d and e) with PM$_{2.5}$ mass concentrations, and MAE$_{365}$ in different RH ranges (panel f).

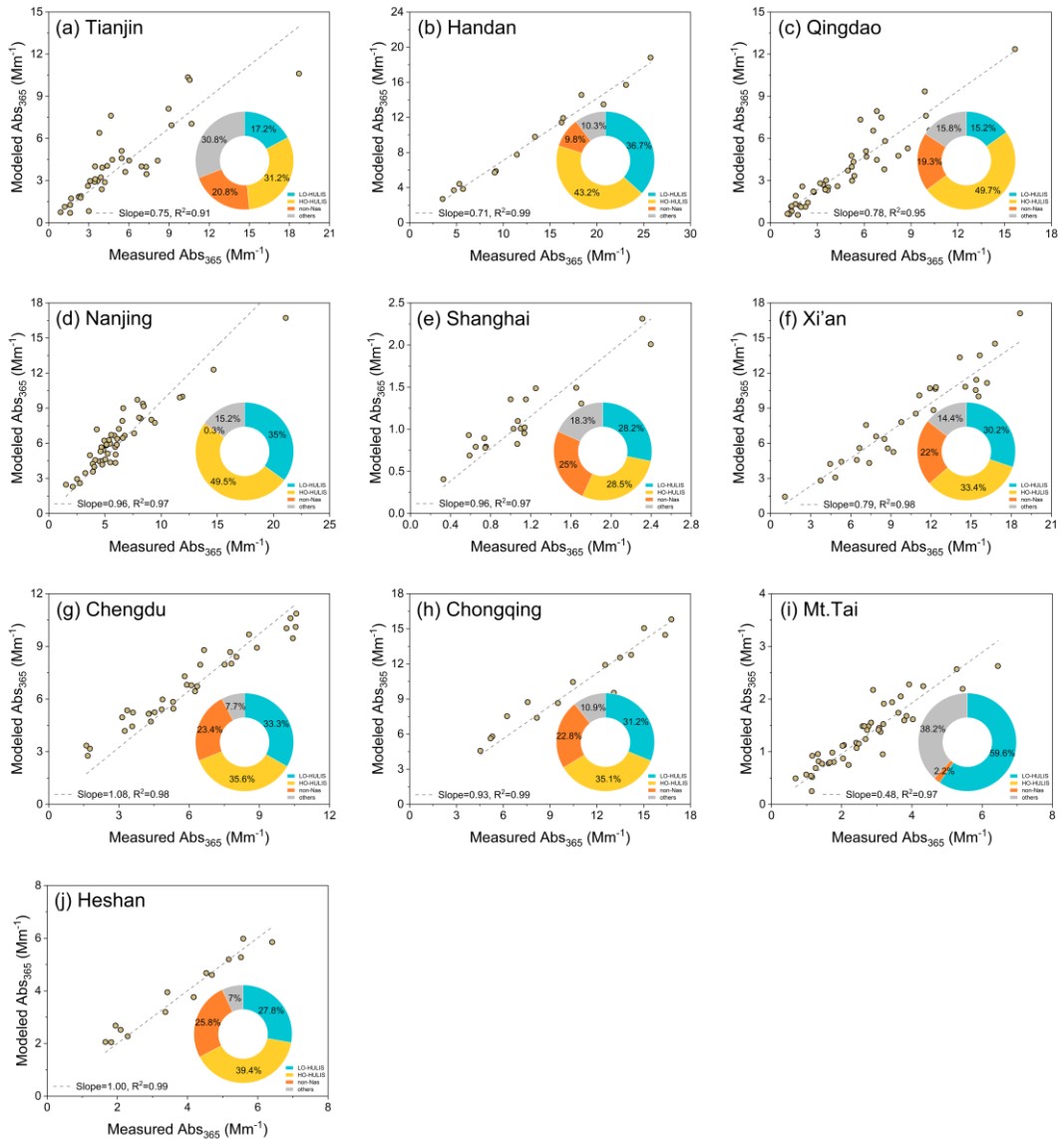

**Figure 8.** Relationships between measured and modeled Abs$_{365}$ based on multiple linear regression (MLR) analysis and fractional contributions of different fluorophores ($F_{max}$) to total Abs$_{365}$ of WSOC (circular graph) at each site.

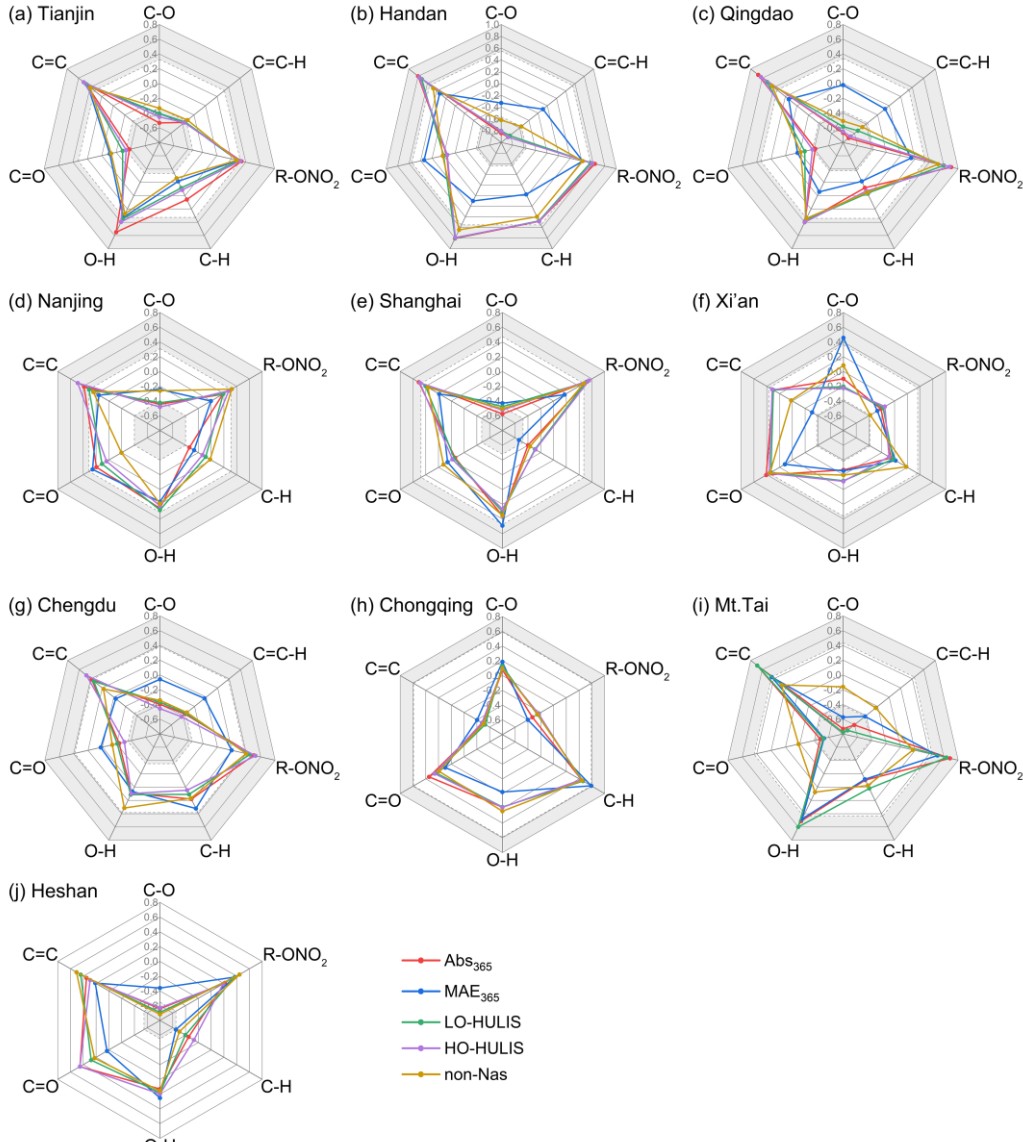

**Figure 9.** Correlation plots among proportion of functional groups, absorbance parameters (Abs$_{365}$ and MAE$_{365}$), and F$_{max}$ of fluorophores (LO-HULIS, HO-HULIS and non-Nas). Note: The shaded in the radar chart denote the significantly positive or negative correlation ($p < 0.05$).