# Peer review of "Measurement Report: Optical and structural properties of atmospheric water-soluble organic carbon in China: Insights from multi-site spectroscopic measurements"

_EGUsphere, 2024_

## Referee Comment (RC2)

**General Comments**: Brown carbon (BrC) is an important constituent of carbonaceous aerosols and significantly contributes to total solar light absorption of aerosols. The manuscript titled "Optical and structural properties of atmospheric water-soluble organic carbon in China: Insights from multi-site spectroscopic measurements" presents measurements of optical properties and structural characteristics of WSOC based on different spectroscopic techniques (absorbance, florescence and FTIR) from different regions of China. Overall, the study promotes a better understanding of the spatial heterogeneity of optical and structural properties of WSOC and their influencing factors (emission sources, aging processes, relative humidity (RH), etc.) in China and deepened the understanding of the contribution of WSOC fluorescence to its light absorption. However, the manuscript has many shortcomings in its current version. It needs through language editing and clarifications at many places throughout the manuscript. It also misses consistency while using different terminologies for the same parameter (for example, authors have used WSOC and WS-BrC interchangeably to refer to BrC). Yet, the study has relevance to the atmospheric research community and can be accepted for publication in the journal after major revision. The detailed comments are given below:

**Major comments**

1. Methodology and elsewhere: What do you mean by "regional site (rs)" in your manuscript? Do you mean "remote/rural site"? It's confusing. Clarify.

2. Methodology (section 2.3): Equation S9 in text S3 is incorrect (could be a typo error). The equation should consist both mass scattering efficiency (MSE) and MAE. Recheck and correct it.

3. Methodology (section 2.3, text S4) and section 3.2 of results section: The author measured WSOC absorbance from 250-700 nm and used WSOC absorbance from 250 to 400 nm in PMF model for source apportionment. However, it is well known that WSOC absorbance < 340 nm is highly influenced by absorbance from nitrate aerosols. Did authors consider this aspect during PMF run? How this will impact the findings?

4. Results and Discussion (section 3.1): The authors observed a significant spatial variability in WSOC, OC, EC, etc. across ten sites. What could be the potential reasons (e.g., different sources, metrology, etc.) behind this variability, discuss briefly?

5. Results and Discussion (section 3.1): The authors compared Abs365 and MAE365 values between different regions (e.g., northwest China, southwest China, etc.). Did you carry out any significance test to check whether difference was significant or not?

6. Results and Discussion (section 3.1): The sampling durations were different at different sites representing different administrative regions (Table S1). Do you think the "day versus night variability" in optical properties could have also contributed to the inter-regional variability observed in optical properties of WS-BrC in your study.

7. Lines 208-210: The authors reported that light absorbing ability (SFE) of WS-BrC and mass concentration of WSOC are directly proportional (related). How did authors come with such conclusions? Please cite relevant studies in this context.

8. Fig. 6c-d: Is this the integrated absorbance from 250-400 nm, clarify?

9. Line 341-346: Please revisit this portion, especially, portion where authors mentioned that photochemical bleaching will be higher during severe pollution days. In fact, opposite is likely to be true as lower pollution levels mean higher visibility, resulting in higher availability of solar flux.

10. The study only discusses optical and structural properties of WS-BrC, which represents only 50-70% fraction of OC. What about water-insoluble OC? What are the implications of your findings on light-absorbing water-insoluble OC?

**Minor Comments**

1. Line 30-34: "The light absorption factors…..impact on fluorophores." Confusing sentence. Rewrite it.

2. Line 45-46: "BrC contributes up to 72% of the total light absorption of aerosols at 370 nm and the direct radiative effect of BrC (+0.048 W·m-2) is about 30% of black carbon (+0.17 W·m-2)". Is this global average or only valid for China? Please clarify.

3. Line 51: Use either "commonly" or "widely". One of them is redundant.

4. Line 51-53: Make it two sentences.

5. Line 55-59: Grammatical mistakes at many places. Some sentences are confusing. Revise it.

6. Line 61: Should be "Spectroscopy-based studies conducted…"

7. Line 81: "methods are often used separately in previous studies"?

8. Line 82-86: Difficult to follow as this is a very long sentence. I suggest to break it into smaller sentences.

9. Line 110: should be "0.45 μm pore-size PTFE syringe filter".

10. Line 118-120: Break it into two sentences like "….calculated. More details can be found in Text S2."

11. Line 152: "Additionally, the XGBoost model is also used to.." "also" is redundant in this sentence.

12. Line 169: "….mass concentrations of carbonaceous components at HS site are not that low…" Compared to what, clarify?

13. Line 157-158: "During the wintertime observation period, WSOC mass concentrations exhibit a significant spatial variation across the ten sites ($p < 0.05$) (see Figure 1 and Table S2)." Which test did you use to derive significance level?

14. Line 308-309: The sentence is confusing. Rewrite it.

15. Line 341: Change "great" to "large".

16. Line 378:Typo?  "WOSC" should be "WSOC"

17. "In contrast, the relationships between MAE365 values and functional groups may differ from Abs365. Similarly, C=C, O-H and R-ONO2 exhibit the strongest correlations with MAE365 at most sites (e.g., TJ, QD, SH, TS, and HS)." Similar to what? These sentences are confusing. Rewrite them.

18. Line 413: "discrepancies" doesn't seem to be the write word here. Replace it.

---

## Author Response (AR2)

**Response to Comments by Review #1**

**Manuscript**: egusphere-2024-2416

**Title**: Measurement Report: Optical and structural properties of atmospheric water-soluble organic carbon in China: Insights from multi-site spectroscopic measurements

**Authors:** Haibiao Chen et al.

**Corresponding author:** Caiqing Yan (email: cyan0325@sdu.edu.cn)

**General Comments:** *The reviewer appreciates the efforts of authors for collecting a good number samples from ten sites of China, and doing a variety of analyses using absorbance and fluorescence spectrometry as well as FTIR. The data have been plotted properly. However, the interpretations are often hand wavy and lack scientific rigour. It limits the sufficient exploitation of such hard-earned data by the authors.*
*The reviewer recommends the major revisions before the manuscript can be considered for publication in the journal ACP.*

**Response to General Comments:** We thank the reviewer for your overall supportive comments. We also appreciate the considerable efforts you put into reviewing the manuscript and providing valuable comments and suggestions for the improvements and clarifications. Based on your comments and suggestions, the manuscript is thoroughly revised. Especially, we have made careful thinking and revisions to interpret the data more rationally, so as to improve the rigor and scientific nature of the discussion.

Below, we detail our responses and resulting edits to all the comments. These are organized such that we first list the review comments in italics and blue, immediately followed by our responses in normal font. To make it clear, the contents in the revised manuscript are presented in quotes and italics, while the newly added contents in the revised manuscript are underlined.

**Major comments:**

**Comment #1:** *L51-53: This is a wrong statement. WSOC is a largely variable fraction of aerosols and numerous studies have shown that water-insoluble OC can be more absorbing BrC (Cheng et al., 2016, Atmospheric Environment). A proper literature review must be added here with the clear information on both water-soluble and water-insoluble BrC. Subsequently, add the limitation of this study which used water-soluble BrC only.*

**Response to Comment #1:** We thank the reviewer for pointing this out and making constructive suggestions. We agree with the reviewer that the statement here (especially that WSOC generally acts as a substitute of BrC) is not

accurate. According to the reviewer's suggestion, we have added a description of water-soluble and water-insoluble BrC in the revised manuscript, and added the limitations that this study only focuses on water-soluble BrC in the conclusion sections. The sentences have been revised as follows,

**(1) In the introduction section:**

*"Solvent-soluble organic carbon (e.g., water-soluble organic carbon, WSOC; methanol-soluble organic carbon, MSOC) is often used to act as a substitute of BrC. In particular, light absorption of WSOC has been extensively studied, due to its widespread presence and high atmospheric abundance in the atmosphere, as well as mature extraction methods, although some previous studies have indicated that water-insoluble OC (WISOC) contains more light-absorbing BrC (Cao et al., 2021; Chen et al., 2024b; Cheng et al., 2016; Yan et al., 2017). Absorption and fluorescence spectroscopy are two of the most widely used methods to reveal optical properties of WSOC (Wang et al., 2022b; Wu et al., 2021). By light-absorption spectroscopy analysis, light absorption characteristics and capabilities of WSOC from different sources or environments are usually characterized by the absorption coefficient or mass absorption efficiency of a specific wavelength over the range of 360-370 nm (average 365 nm) (Hecobian et al., 2010). And the direct radiative forcing of WSOC can be further estimated by simplified radiative forcing models combined with the measured absorption coefficient."*

**(2) In the conclusion section:**

*"Additionally, it is important to note that this study only focused on WSOC. Since the WISOC may have stronger light absorption capacity, further research on light absorption, composition and structure of WISOC (especially in northern China in winter) and the correlation between them are also needed in the future."*

**References:**

Cao, T., Li, M., Zou, C., Fan, X., Song, J., Jia, W., Yu, C., Yu, Z., and Peng, P. a.: Chemical composition, optical properties, and oxidative potential of water- and methanol-soluble organic compounds emitted from the combustion of biomass materials and coal, Atmospheric Chemistry and Physics, 21, 13187-13205, https://doi.org/10.5194/acp-21-13187-2021, 2021.

Chen, H., Zhou, R., Fang, L., Sun, H., Yang, Q., Niu, H., Liu, J., Tian, Y., Cui, M., and Yan, C.: Variations in optical properties of water- and methanol-soluble organic carbon in $PM_{2.5}$ in Tianjin and Handan over the wintertime of 2018-2020, Atmospheric Research, 303, 107332, https://doi.org/10.1016/j.atmosres.2024.107332, 2024b.

Cheng, Y., He, K. B., Du, Z. Y., Engling, G., Liu, J. M., Ma, Y. L., Zheng, M., and Weber, R. J.: The characteristics of brown carbon aerosol during winter in Beijing, Atmospheric Environment, 127, 355-364,

https://doi.org/10.1016/j.atmosenv.2015.12.035, 2016.

Yan, C., Zheng, M., Bosch, C., Andersson, A., Desyaterik, Y., Sullivan, A.P., Collett, J.L., Zhao, B., Wang, S.X.,
60      He, K.B., Gustafsson, Ö. Important fossil source contribution to brown carbon in Beijing during winter. Scientific
Reports, 7, 43182, https://doi.org/10.1038/srep43182, 2017.

**Comment #2:** *Section 2.1: Briefly provide details of sampler, sampling duration, frequency, total number of samples, etc and then refer Table S1 for details.*

**Response to Comment #2:** We thank the reviewer's kind reminder and constructive suggestions. We have taken
65      the reviewer's suggestion, added a brief description of the sampling information, and made the following modifications in the revised manuscript,

*"PM$_{2.5}$ samples are collected at eight urban sites and two regional sites in China during the late November and January of 2019-2020 (see Figure S1 and Table S1). .... In this study, daytime, nighttime or daily PM$_{2.5}$ samples were collected using medium- or high-volume samplers at different sites with a sampling duration of 11 h-24 h for*
70      *each sample. More detailed sampling information and sample sizes are summarized in Table S1 in the Supplement. It is worth noting that in this study, the daily average of the parameters measured at each site is used for subsequent summary and comparison."*

**Comment #3:** *L119-120: Give a valid reason for selecting this wavelength range only for AAE calc*

**Response to Comment #3:** We appreciate the reviewer's kind reminder and apologize for the confusion caused.
75      In this study, the selection of wavelength range was mainly based on the following considerations: (1) to avoid the light absorption interference of inorganic substances (such as ammonium nitrate, sodium nitrate and nitrate ions) at shorter wavelengths (< 250 nm) (Afsana et al., 2022), (2) to ensure that WSOC has a more significant light absorption signal (see the new Figure. 4 in the revised manuscript), and (3) the fitting of the power-law of log Abs$_\lambda$ and log $\lambda$ used to calculate the AAE values is good (see Figure R1). In view of these above criteria, we re-select
80      the data in the range of 300-500 nm to calculate the AAE values in the revised manuscript.

In the revised manuscript, we have added the reason for selecting the wavelength range for the AAE calculation to the supplementary materials (please see Text S2), and revised the relevant discussion and description of AAE values in the main text and chart accordingly.

*"In this study, AAE is fitted over the wavelength range of 300-500 nm in consideration of avoiding the interference*

85    *from light-absorbing inorganic compounds (e.g., ammonium nitrate, sodium nitrate and nitrate ions) at shorter wavelengths (< 250 nm) and ensuring significant light absorption signals of WSOC at longer wavelengths (Afsana et al., 2022; Ting et al., 2022; Yan et al., 2015). Moreover, the power-law fit of all samples' absorption coefficients between 300 and 500 nm is good with $r^2 > 0.99$ (see Figure S2).* " (in the Supplement)

[Figure]

90    **Figure 4.** *(a) Light absorption coefficient spectrum of WSOC at each site and the average light absorption spectrum in (b) north China, (c) east China, (d) northwest China, (e) southwest China, and (f) regional sites. Note: The color bar represents the magnitude of the $Abs_\lambda$.*

[Figure]

**Figure R1.** Comparison of power-law fitting curves for light absorption in different wavelength ranges: (a) 300-500 nm and (b) 300-600 nm.

[Figure]

*Figure S2. The power-law fitting curves of light absorption in the wavelength range of 300-500 nm.*

**References:**

Afsana, S., Zhou, R., Miyazaki, Y., Tachibana, E., Deshmukh, D. K., Kawamura, K., and Mochida, M.: Abundance,

100  chemical structure, and light absorption properties of humic-like substances (HULIS) and other organic fractions of forest aerosols in Hokkaido, Sci Rep, 12, 14379, https://doi.org/10.1038/s41598-022-18201-z, 2022.

Ting, Y.-C., Ko, Y.-R., Huang, C.-H., Cheng, Y.-H., and Huang, C.-H.: Optical properties and potential sources of water-soluble and methanol-soluble organic aerosols in Taipei, Taiwan, Atmospheric Environment, 290, 119364, https://doi.org/10.1016/j.atmosenv.2022.119364, 2022.

105  Yan, C. Q., Zheng, M., Sullivan, A. P., Bosch, C., Desyaterik, Y., Andersson, A., Li, X. Y., Guo, X. S., Zhou, T., Gustafsson, O., and Collett, J. L.: Chemical characteristics and light-absorbing property of water-soluble organic carbon in Beijing: Biomass burning contributions, Atmospheric Environment, 121, 4-12, https://doi.org/10.1016/j.atmosenv.2015.05.005, 2015.

**Comment #4:** *L165-170: This is very hand wavy discussion. Statement shall be substantiated by logical arguments* 110  *and supporting data.*

**Response to Comment #4:** We agree with the reviewer that the statement should be substantiated by logical arguments and supporting data. To make the discussion here more reasonable, we used Hybrid Single-Particle Lagrangian Integrated Trajectory (HYSPLIT) model to analyze the 48-h backward air mass trajectories that arriving at the ten sites to assess the impact of anthropogenic and oceanic emissions. And this statement has been revised 115  as follows,

*"Moreover, concentrations of carbonaceous components in inland cities (i.e., HD, NJ, XA, CD, CQ) are much higher than those in coastal cities (i.e., TJ, QD, SH) (p < 0.01), which is consistent with that reported in previous studies that have shown that air masses from the ocean generally contain lower levels of aerosol content and carbonaceous components (Chen et al., 2023; Diesch et al., 2012; Mo et al., 2022; Zhang et al., 2022b). The 48-h* 120  *backward air mass trajectory analysis shows that about 1/5 to 1/2 of the air masses that arrive at the coastal cities during the observation period pass through the ocean region, while the inland cities are predominately affected by continental air masses, which may contain a large number of anthropogenic aerosols (see Figure S4). Furthermore, it is worth noting that the regional site TS in NCP has a relatively low mass concentration of carbonaceous components compared to urban sites, which may be due to its high altitude (~1500 m) and low local anthropogenic* 125  *activities (Jiang et al., 2020). In contrast, the mass concentrations of carbonaceous components at HS (another regional site) in the PRD region are relatively higher compared to TS site. The backward air mass trajectory analysis indicates that more than 80% of the air masses arriving at the HS site originated from the PRD region and are accompanied by low wind speeds (1.54 m·s$^{-1}$ on average during the sampling period). This suggests that*

*there may be high anthropogenic emissions in the PRD region during the winter sampling period."*

[Figure]

**Figure S4.** *Clusters of air masses derived from backward trajectory analysis at the ten sites. The 48-h backward trajectories at each site are calculated every 1 h and clustered at an ending height of 500 m above ground level based on the MeteoInfoMap software.*

**References:**

Chen, H., Yan, C., Fu, Q., Wang, X., Tang, J., Jiang, B., Sun, H., Luan, T., Yang, Q., Zhao, Q., Li, J., Zhang, G., Zheng, M., Zhou, X., Chen, B., Du, L., Zhou, R., Zhou, T., and Xue, L.: Optical properties and molecular composition of wintertime atmospheric water-soluble organic carbon in different coastal cities of eastern China,

Science of the Total Environment, 892, 164702, https://doi.org/10.1016/j.scitotenv.2023.164702, 2023.

Diesch, J. M., Drewnick, F., Zorn, S. R., von der Weiden-Reinmüller, S. L., Martinez, M., and Borrmann, S.: Variability of aerosol, gaseous pollutants and meteorological characteristics associated with changes in air mass origin at the SW Atlantic coast of Iberia, Atmospheric Chemistry and Physics, 12, 3761-3782, https://doi.org/10.5194/acp-12-3761-2012, 2012.

Jiang, Y., Xue, L., Gu, R., Jia, M., Zhang, Y., Wen, L., Zheng, P., Chen, T., Li, H., Shan, Y., Zhao, Y., Guo, Z., Bi, Y., Liu, H., Ding, A., Zhang, Q., and Wang, W.: Sources of nitrous acid (HONO) in the upper boundary layer and lower free troposphere of the North China Plain: Insights from the Mount Tai Observatory, Atmospheric Chemistry and Physics, 20, 12115-12131, https://doi.org/10.5194/acp-20-12115-2020, 2020.

Mo, Y. Z., Zhong, G. C., Li, J., Liu, X., Jiang, H. X., Tang, J., Jiang, B., Liao, Y. H., Cheng, Z. N., and Zhang, G.: The sources, molecular compositions, and light absorption properties of water-soluble organic carbon in marine aerosols from South China Sea to the Eastern Indian Ocean, Journal of Geophysical Research: Atmospheres, 127, https://doi.org/10.1029/2021JD036168, 2022.

Zhang, J., Qi, A., Wang, Q., Huang, Q., Yao, S., Li, J., Yu, H., and Yang, L.: Characteristics of water-soluble organic carbon (WSOC) in $PM_{2.5}$ in inland and coastal cities, China, Atmos. Pollut. Res., 13, 101447, https://doi.org/10.1016/j.apr.2022.101447, 2022b.

**Comment #5:** *L190-191: This discussion should also consider the effects of meteorological processes on $Abs_{365}$ e.g., photo-bleaching or photo-enhancement.*

**Response to Comment #5:** We agree with the reviewer that atmospheric processes, including meteorological processes, also have an impact on $Abs_{365}$. To make it more accurate, we have made the following modifications,

*"This spatial variation may be related to the diversity of WSOC sources and can be affected by the atmospheric processes (including meteorological processes), and is intrinsically related to the chemical composition and structures of WSOC at different sites (Wang et al., 2023; Wang et al., 2024). Previous studies have indicated that the increase in primary emissions such as coal combustion and biomass burning during the winter heating period in Northern China will lead to an enhancement of the WSOC light absorption (Yan et al., 2017; Zhang et al., 2021). The strong correlation between $Abs_{365}$ and POC (r range: 0.59-0.90) or SOC (r range: 0.43-0.97) (p < 0.01) (see Figure S6) indicates that light-absorbing components in WSOC are simultaneously affected by both primary emission and secondary formation. The effects of different factors such as chemical composition/structure, and*

*meteorological conditions on the light absorption of WSOC will be discussed in detail in the following sections."*

**References:**

Wang, D., Shen, Z., Yang, X., Huang, S., Luo, Y., Bai, G., and Cao, J.: Insight into the Role of $NH_3/NH_4^+$ and $NO_x/NO_3^-$ in the Formation of Nitrogen-Containing Brown Carbon in Chinese Megacities, Environmental Science & Technology, https://doi.org/10.1021/acs.est.3c10374, 2024.

Wang, Y., Feng, Z., Yuan, Q., Shang, D., Fang, Y., Guo, S., Wu, Z., Zhang, C., Gao, Y., Yao, X., Gao, H., and Hu, M.: Environmental factors driving the formation of water-soluble organic aerosols: A comparative study under contrasting atmospheric conditions, Science of The Total Environment, 866, https://doi.org/10.1016/j.scitotenv.2022.161364, 2023.

Yan, C., Zheng, M., Bosch, C., Andersson, A., Desyaterik, Y., Sullivan, A.P., Collett, J.L., Zhao, B., Wang, S.X., He, K.B., Gustafsson, Ö. Important fossil source contribution to brown carbon in Beijing during winter. Scientific Reports, 7, 43182. https://doi.org/10.1038/srep43182, 2017.

**Comment #6:** *L198-204: These statements shall be endorsed by some source apportionment studies which reported biomass/residential heating as major source over these regions.*

**Response to Comment #6:** We thank the reviewer's constructive suggestion. We take the reviewer's suggestion and have supplemented the statements as follows,

*"Figure 1 illustrates the $log_{10}(MAE_{405})$ and $AAE_{300-500}$ values of WSOC measured in this study and reported in previous studies. The map in $log_{10}(MAE_{405})$ vs. $AAE_{300-500}$ space has been proposed by Saleh (2020) to classify BrC types based on light-absorbing properties. Notably, most values measured at the ten sites in this study fall in the regions of weakly-absorbing BrC (W-BrC), which are similar to the values of ambient samples reported in previous studies. However, there are slight differences in the distribution of values between each site. QD has the widest range of $log_{10}(MAE_{405})$ and $AAE_{300-500}$ values, and is also close to the values of biomass burning samples. Previous studies have indicated that biomass burning (especially those related to residential heating and cooking activities) around the sampling site in QD in winter is a major source contributor to atmospheric particulate matter (Li et al., 2024), and has a significant impact on the light absorption of WSOC (Zhan et al., 2022a; Zhan et al., 2022b). The difference in light absorption of WSOC generated from different fuel (e.g., corn straw, rice straw and pine branch) combustion may be the reason for the wide range of $log_{10}(MAE_{405})$ and $AAE_{300-500}$ values in QD (Fan et al., 2016). The values in SH are much closer to very weakly-absorbing BrC (VW-BrC) compared to other sites, which may be*

*related to the influence of marine air mass, as the backward air mass trajectory analysis shows that more than half*

195    *of the air masses arriving in SH passing through the ocean area during the sampling period (see Figure S4). In*

*contrast, the values in XA are mainly distributed in the moderately-absorbing BrC (M-BrC) region, with a few*

*samples falling in the W-BrC region. This indicates that WSOC in XA has a stronger light-absorbing capacity,*

*which is consistent with its higher $Abs_{365}$ and $MAE_{365}$ values. The average value in XA is much closer to that of coal*

*combustion source samples. Previous studies conducted in XA have shown that the light-absorbing capacity of*

200    *WSOC in XA region is always at a high level, especially during the winter heating period when anthropogenic*

*emissions such as biomass burning and coal combustion activities increased (Huang et al., 2018; Lei et al., 2019;*

*Yuan et al., 2020). The cause of the high light absorption level of WSOC in XA needs to be further investigated."*

[Figure]

**Figure 1.** *Graphical representation of optical-based BrC classes in $\log_{10}(MAE_{405})$-AAE space. The shaded regions*

205    *represent very weakly light-absorbing BrC (VW-BrC), weakly light-absorbing BrC (W-BrC), moderately light-*

*absorbing BrC (M-BrC), strongly light-absorbing BrC (S-BrC), and absorbing BC, respectively.*

**References:**

Fan, X. J., Wei, S. Y., Zhu, M. B., Song, J. Z., and Peng, P. A.: Comprehensive characterization of humic-like

substances in smoke $PM_{2.5}$ emitted from the combustion of biomass materials and fossil fuels, Atmospheric

210    Chemistry and Physics, 16, 13321-13340, https://doi.org/10.5194/acp-16-13321-2016, 2016.

Huang, R. J., Yang, L., Cao, J., Chen, Y., Chen, Q., Li, Y., Duan, J., Zhu, C., Dai, W., Wang, K., Lin, C., Ni, H.,

Corbin, J. C., Wu, Y., Zhang, R., Tie, X., Hoffmann, T., O'Dowd, C., and Dusek, U.: Brown carbon aerosol in urban Xi'an, northwest China: The composition and light absorption properties, Environmental Science & Technology, 52, 6825-6833, https://doi.org/10.1021/acs.est.8b02386, 2018.

215 Lei, Y., Shen, Z., Zhang, T., Lu, D., Zeng, Y., Zhang, Q., Xu, H., Bei, N., Wang, X., and Cao, J.: High time resolution observation of $PM_{2.5}$ Brown carbon over Xi'an in northwestern China: Seasonal variation and source apportionment, Chemosphere, 237, 124530, https://doi.org/10.1016/j.chemosphere.2019.124530, 2019.

Li, R.Y., Yan, C.Q., Meng, Q.P., Yue, Y., Jiang, W., Yang, L.X., Zhu, Y.J., Xue, L.K., Gao, S.P., Liu, W.J., Chen, T.X., Meng, J.J. Key toxic components and sources affecting oxidative potential of atmospheric particulate matter
220 using interpretable machine learning: Insights from fog episodes. Journal of Hazardous Materials, 465, 133175, https://doi.org/10.1016/j.jhazmat.2023.133175, 2024.

Saleh, R.: From measurements to models: Toward accurate representation of brown carbon in climate calculations, Current Pollution Reports, 6, 90-104, https://doi.org/10.1007/s40726-020-00139-3, 2020.

[revised manuscript text omitted]

**Comment #8:** *L249-254: The discussion is mostly data description. Author shall attempt to extract science and more substance out of it.*

**Response to Comment #8:** We thank the reviewer for pointing out this problem and putting forward constructive suggestions. We have reanalyzed these data and made the following changes in the revised manuscript,

*"Figure 2 illustrates the average fluorescence volume (FV) and relative contributions of different fluorophores at different sites.* Clearly, the average FV of all fluorophores varies in the range of $2.50 \times 10^2$-$9.76 \times 10^3$ RU-nm$^2$, showing great spatial variability. Overall, the LO-HULIS fluorophore has the highest FV ($1.49 \times 10^3 \pm 4.74 \times 10^2$-$9.76 \times 10^3 \pm 5.82 \times 10^3$ RU-nm$^2$ on average) and accounts for the largest proportion ($42.2 \pm 5.59\%$-$60.3 \pm 2.11\%$ on average) in the total fluorophores at most sites (except for TS, NJ and CD), demonstrating the widespread existence of combustion-related sources (especially biomass burning) and their important influence on the fluorophore. In contrast, HO-HULIS fluorophore accounts for the lowest proportion ($12.5 \pm 1.32\%$-$23.8 \pm 2.43\%$ on average, except for TJ and TS). This suggests that the effect of secondary anthropogenic sources at different location may be relatively small during the winter study period. The relatively high FV values of LO-HULIS and non-Nas in northern cities ($1.62 \times 10^6$ RU-nm$^2$ in total) compared to southern cities ($4.37 \times 10^5$ RU-nm$^2$ in total) and regionals sites ($1.76 \times 10^5$ RU-nm$^2$ in total) further indicate the impact of increased primary emissions during the heating season, especially in northern China (Cao et al., 2023; Li et al., 2023b). *This is consistent with a previous study by Cao et al., (2024a), which* showed that the FV of fluorophores in BrC during the winter heating season was significantly higher than that during the non-heating season.*"*

**References:**

Cao, J., Shang, J., Kuang, Y., Jiang, X., Shi, X., and Qiu, X.: Molecular composition of Beijing PM$_{2.5}$ Brown carbon associated with fluorescence revealed by gas chromatography time-of-flight mass spectrometry and parallel factor analysis, Atmospheric Environment, 333, https://doi.org/10.1016/j.atmosenv.2024.120670, 2024a.

Cao, T., Li, M., Xu, C., Song, J., Fan, X., Li, J., Jia, W., and Peng, P. a.: Technical note: Chemical composition and source identification of fluorescent components in atmospheric water-soluble brown carbon by excitation-emission matrix spectroscopy with parallel factor analysis-potential limitations and applications, Atmospheric Chemistry and Physics, 23, 2613-2625, https://doi.org/10.5194/acp-23-2613-2023, 2023.

Li, P., Yue, S., Yang, X., Liu, D., Zhang, Q., Hu, W., Hou, S., Zhao, W., Ren, H., Li, G., Gao, Y., Deng, J., Xie, Q., Sun, Y., Wang, Z., and Fu, P.: Fluorescence properties and chemical composition of fine particles in the background atmosphere of North China, Advances in Atmospheric Sciences, 40, 1159-1174, https://doi.org/10.1007/s00376-022-2208-x, 2023b.

**Comment #9:** *L253-256: Use proper significant digits while reporting the numbers throughout the MS. In the number 8436.99, value after decimal (0.99) has no meaning when the SD is also in several thousands.*

**Response to Comment #9:** We thank the reviewer's kind reminder. We have checked the figures throughout the manuscript and tried to unify the significant digits in the main text and the tables (Table 1 and Table S2). For example, for data greater than 1 (we use the scientific notation for numbers greater than 100), we have uniformly retained three significant digits. As for numbers less than 1, we have kept at most two decimal places. The specific modifications in the main text and tables are as follows,

(1) *"...vary from 2.69 to 19.5 (4.58 ± 0.93-10.1 ± 2.69 on average) and 22.7% to 96.1% (53.4 ± 4.94%-73.3 ± 10.0% on average) across the ten sites…"*

(2) *"As shown in Table 1 and Figure S1, the average $Abs_{365}$ (1.12 ± 0.53-13.1 ± 6.95 $Mm^{-1}$) and $MAE_{365}$ (0.56 ± 0.11-1.26 ± 0.34 $m^2 \cdot g^{-1}$) of WSOC at the ten sites display significant spatial discrepancies ($p < 0.05$), with HD (SH) has the highest (lowest) average $Abs_{365}$ (13.1 ± 6.95 $Mm^{-1}$ (1.12 ± 0.53 $Mm^{-1}$)) and $MAE_{365}$ (1.26 ± 0.34 $m^2 \cdot g^{-1}$ (0.56 ± 0.11 $m^2 \cdot g^{-1}$)), respectively."*

(3) *"Figure 2 illustrates the average fluorescence volume (FV) and relative contributions of different fluorophores at different sites. Clearly, the average FV of all fluorophores varies in the range of $2.50 \times 10^2$-$9.76 \times 10^3$ $RU$-$nm^2$, showing great spatial variability. Overall, the LO-HULIS fluorophore has the highest FV ($1.49 \times 10^3 \pm 4.74 \times 10^2$-$9.76 \times 10^3 \pm 5.82 \times 10^3$ $RU$-$nm^2$ on average) and accounts for the largest proportion (42.2 ± 5.59%-60.3 ± 2.11% on average) in the total fluorophores at most sites (except for TS, NJ and CD), demonstrating the widespread existence of combustion-related sources (especially biomass burning) and their important influence on the fluorophore. In contrast, HO-HULIS fluorophore accounts for the lowest proportion (12.5 ± 1.32%-23.8 ± 2.43% on average, except for TJ and TS). This suggests that the effect of secondary anthropogenic sources at different location may be relatively small during the winter study period. The relatively high FV values of LO-HULIS and non-Nas in northern cities ($1.62 \times 10^6$ $RU$-$nm^2$ in total) compared to southern cities ($4.37 \times 10^5$ $RU$-$nm^2$ in total) and regionals sites ($1.76 \times 10^5$ $RU$-$nm^2$ in total) further indicate the impact of increased primary emissions during the heating season, especially in northern China."*

(4) *"**Table 1.** Light absorption and fluorescence parameters of WSOC measured in this study."*

| Sites | Regions | $Abs_{365}$ ($Mm^{-1}$) | $MAE_{365}$ ($m^2 \cdot g^{-1}$) | $AAE_{300-500}$ | FI | BIX | HIX |
|---|---|---|---|---|---|---|---|
| | | Avg ± SD | Avg ± SD | Avg ± SD | Avg ± SD | Avg ± SD | Avg ± SD |
| Tianjin (TJ) | North China | 5.57 ± 3.83 | 0.89 ± 0.22 | 6.58 ± 0.74 | 1.48 ± 0.06 | 0.97 ± 0.06 | 2.91 ± 0.37 |
| Handan | | 13.1 ± 6.95 | 1.26 ± 0.34 | 5.96 ± 0.36 | 1.55 ± 0.08 | 1.00 ± 0.11 | 1.07 ± 0.43 |

| | (HD) | | | | | | |
|---|---|---|---|---|---|---|---|
| Qingdao (QD) | East China | 4.80 ± 3.09 | 1.03 ± 0.34 | 7.04 ± 0.89 | 1.58 ± 0.09 | 1.08 ± 0.11 | 1.69 ± 0.32 |
| Nanjing (NJ) | East China | 6.26 ± 3.26 | 0.89 ± 0.25 | 6.66 ± 0.35 | 1.49 ± 0.17 | 0.82 ± 0.12 | 0.56 ± 0.20 |
| Shanghai (SH) | | 1.12 ± 0.53 | 0.56 ± 0.11 | 7.02 ± 0.54 | 1.57 ± 0.09 | 1.02 ± 0.08 | 1.98 ± 0.26 |
| Xi'an (XA) | Northwest China | 10.6 ± 4.42 | 1.04 ± 0.11 | 4.83 ± 1.01 | 1.58 ± 0.08 | 0.96 ± 0.09 | 1.53 ± 0.33 |
| Chengdu (CD) | Southwest China | 5.99 ± 2.61 | 0.65 ± 0.11 | 6.24 ± 0.47 | 1.54 ± 0.10 | 0.80 ± 0.07 | 1.18 ± 0.25 |
| Chongqing (CQ) | | 10.6 ± 4.10 | 0.98 ± 0.09 | 5.78 ± 0.39 | 1.51 ± 0.04 | 0.84 ± 0.04 | 1.37 ± 0.24 |
| Mt. Tai (TS) | Regional site | 2.66 ± 1.22 | 0.74 ± 0.24 | 7.11 ± 0.46 | 1.31 ± 0.17 | 0.75 ± 0.15 | 0.26 ± 0.11 |
| Heshan (HS) | | 3.76 ± 1.55 | 0.64 ± 0.12 | 6.28 ± 0.30 | 1.49 ± 0.03 | 0.77 ± 0.06 | 0.91 ± 0.15 |

325     (5) *"Table S2. Mass concentrations and related ratios of carbonaceous components measured in this study."*

| Sites | Regions | OC ($\mu g \cdot m^{-3}$) | EC ($\mu g \cdot m^{-3}$) | WSOC ($\mu g \cdot m^{-3}$) | OC/EC | WSOC/OC (%) | POC ($\mu g \cdot m^{-3}$) | SOC ($\mu g \cdot m^{-3}$) |
|---|---|---|---|---|---|---|---|---|
| | | Avg ± SD | Avg ± SD | Avg ± SD | Avg ± SD | Avg ± SD | Avg ± SD | Avg ± SD |
| Tianjin (TJ) | north China | 10.0 ± 6.06 | 1.20 ± 0.53 | 5.96 ± 3.21 | 7.99 ± 2.08 | 59.9 ± 11.8 | 5.64 ± 2.47 | 4.38 ± 3.94 |
| Handan (HD) | | 17.9 ± 8.56 | 2.44 ± 0.48 | 10.6 ± 5.33 | 7.06 ± 2.52 | 55.2 ± 13.3 | 8.79 ± 1.72 | 9.08 ± 7.34 |
| Qingdao (QD) | east China | 8.69 ± 5.60 | 1.34 ± 0.83 | 4.68 ± 3.01 | 6.45 ± 1.15 | 55.1 ± 11.6 | 5.86 ± 3.62 | 2.83 ± 2.49 |
| Nanjing (NJ) | | 10.6 ± 4.16 | 2.36 ± 0.83 | 6.82 ± 2.00 | 4.58 ± 0.93 | 65.2 ± 11.9 | 7.03 ± 2.48 | 3.61 ± 2.84 |
| Shanghai (SH) | | 3.31 ± 1.48 | 0.35 ± 0.16 | 1.97 ± 0.83 | 10.1 ± 2.69 | 61.0 ± 6.99 | 1.72 ± 0.79 | 1.59 ± 0.94 |
| Xi'an (XA) | northwest China | 17.8 ± 7.24 | 2.86 ± 1.49 | 10.2 ± 4.55 | 6.90 ± 1.65 | 57.3 ± 6.93 | 12.9 ± 6.74 | 4.87 ± 2.37 |
| Chengdu (CD) | southwest China | 12.2 ± 4.48 | 1.87 ± 0.51 | 8.98 ± 3.09 | 6.44 ± 1.41 | 73.3 ± 10.0 | 6.02 ± 1.63 | 6.17 ± 3.22 |
| Chongqing (CQ) | | 19.6 ± 6.48 | 2.25 ± 0.79 | 10.5 ± 3.45 | 9.02 ± 1.94 | 53.4 ± 4.94 | 13.6 ± 4.78 | 5.96 ± 3.98 |
| Mt. Tai (TS) | regional site | 4.17 ± 2.73 | 0.71 ± 0.74 | 3.60 ± 1.22 | 6.97 ± 3.11 | 65.1 ± 19.6 | 1.91 ± 2.00 | 2.26 ± 1.58 |
| Heshan (HS) | | 9.17 ± 3.54 | 1.38 ± 0.31 | 5.85 ± 2.12 | 6.45 ± 1.52 | 64.4 ± 11.9 | 5.37 ± 1.20 | 3.80 ± 2.55 |

**Comment #10:** *L322-23: What could be possible reasons for such difference?*

**Response to Comment #10:** We thank the reviewer for the good question and apologize for the insufficient discussion. In the revised manuscript, this paragraph has been rewritten as we have rerun the PMF model, and we

have added the possible reasons for the difference accordingly.

330    According to the UV-Vis absorption spectra (see the new Figure 4), the WSOC absorption spectra of the ten sites can be roughly divided into two categories: (1) unimodal, that is, the light absorption continues to decline from 250 nm to 700 nm, with a peak around 250 nm; (2) bimodal, that is, there are two significant absorption peaks at around 265 nm and 300 nm. Interestingly, these two types of spectra happen to correspond to the East China sites (i.e., QD, NJ, SH and TS) and the outside East China sites (i.e., TJ, HD, XA, CD, CQ and HS), respectively. Therefore, the

335    WSOC spectra of the two categories were separately put into the PMF model for analysis. In view of the fact that WSOCs at ten sites all have relatively significant light absorption in the range of 250-500 nm, and in order to be consistent with the calculation range of AAE, PMF analysis is carried out for spectra in the band range of 250-500 nm in the revised draft. Therefore, this section has been rewritten in the revised manuscript as follows,

[revised manuscript text omitted]

**Comment #11:** *L335-343: Such discussions shall be supported by some other independently measured species. Otherwise, it will be considered as hypothetical statement.*

405 **Response to Comment #11:** We thank the reviewer's kind reminder and constructive suggestion. Combining with the **Comment #12**, we have rewritten this section in the manuscript as follows,

*"To investigate the influence of air quality levels on the light absorption properties of WSOC, the sampling days are classified into five pollution levels including clean (0-35 $\mu g \cdot m^{-3}$), relatively clean (35-75 $\mu g \cdot m^{-3}$), slightly polluted (75-115 $\mu g \cdot m^{-3}$), moderately polluted (115-150 $\mu g \cdot m^{-3}$), and heavily polluted (> 150 $\mu g \cdot m^{-3}$) according to*

410 *the national ambient air quality daily Grade-II standard threshold values and ambient air quality indices. As shown in Figure 7a and b, $Abs_{365}$ and $MAE_{365}$ of WSOC both increase with the increase of pollution levels, in which $Abs_{365}$ changes significantly (p < 0.01) while $MAE_{365}$ changes relatively gently. The enhancement of WSOC light*

*absorption under high pollution conditions may be related to the increase of WSOC concentration, light absorption capacity and light-absorbing species. Previous studies have reported that the mass fractions of oxidized organic aerosols increase significantly with the increase of $PM_{2.5}$ mass concentration, and the oxidized organic aerosols contain a large number of light-absorbing species such as nitroaromatics compounds (You et al., 2024). In this study, the relative abundances of O-H, C=C and $R-ONO_2$ functional groups, which are related to aromatic compounds and have a good positive correlation with the light absorption of WSOC, increase with $PM_{2.5}$ mass concentration (see Figure S14, and discussion in the next section). Additionally, the accumulation of anthropogenic emissions (especially those sources with strong light-absorbing BrC such as biomass burning and coal combustion sources) at high pollution levels will lead to an increase in BrC chromophore types and overall light absorption capacity (Li et al., 2020a; Tang et al., 2020; Wei et al., 2020)."*

[Figure]

**Figure S14.** *Variations of the relative abundance of functional groups (a) C=C, (b) O-H and (c) $R-ONO_2$ and the proportion of (d) LO-HULIS, (e) HO-HULIS and (f) non-Nas fluorophores with $PM_{2.5}$ mass concentrations.*

**References:**

Li, J. J., Zhang, Q., Wang, G. H., Li, J., Wu, C., Liu, L., Wang, J. Y., Jiang, W. Q., Li, L. J., Ho, K. F., and Cao, J. J.: Optical properties and molecular compositions of water-soluble and water-insoluble brown carbon (BrC) aerosols in northwest China, Atmospheric Chemistry and Physics, 20, 4889-4904, https://doi.org/10.5194/acp-20-4889-2020, 2020a.

Tang, J., Li, J., Su, T., Han, Y., Mo, Y. Z., Jiang, H. X., Cui, M., Jiang, B., Chen, Y. J., Tang, J. H., Song, J. Z., Peng,

P. A., and Zhang, G.: Molecular compositions and optical properties of dissolved brown carbon in biomass burning, coal combustion, and vehicle emission aerosols illuminated by excitation-emission matrix spectroscopy and Fourier transform ion cyclotron resonance mass spectrometry analysis, Atmospheric Chemistry and Physics, 20, 2513-2532, https://doi.org/10.5194/acp-20-2513-2020, 2020.

Wei, Y., Chen, H., Sun, H., Zhang, F., Shang, X., Yao, L., Zheng, H., Li, Q., and Chen, J.: Nocturnal $PM_{2.5}$ explosive growth dominates severe haze in the rural North China Plain, Atmospheric Research, 242, https://doi.org/10.1016/j.atmosres.2020.105020, 2020.

You, B., Zhang, Z., Du, A., Li, Y., Sun, J., Li, Z., Chen, C., Zhou, W., Xu, W., Lei, L., Fu, P., Hou, S., Li, P., and Sun, Y.: Seasonal characterization of chemical and optical properties of water-soluble organic aerosol in Beijing, Science of The Total Environment, 930, 172508, https://doi.org/10.1016/j.scitotenv.2024.172508, 2024.

**Comment #12:** *L345-346: It looks a wrong statement. The mentioned processes make the light absorption properties of BrC variable, as reported in numerous studies.*

**Response to Comment #12:** We apologize for the misleading. Taking into account the **Comment #11** by the reviewer, we have rewritten this section. Please refer to the **Response to Comment #11**.

**Comment #13:** *L350-351: This statement looks counter intuitive.*

**Response to Comment #13:** We thank the reviewer for pointing this out. Combining with the **Comment #14**, we have revised this statement in the revised manuscript as follows,

*"The fluorescence volumes normalized (NFV) by WSOC concentration of different fluorophores exhibit different variation trends with $PM_{2.5}$ mass concentrations (see Figure 7c-e). Overall, the total NFV value of HULIS increases with $PM_{2.5}$ concentrations, with the increase of HO-HULIS being more monotonous and significant ($p < 0.01$, based on Spearman's rank correlation test) while the increase of NFV of LO-HULIS being less significant ($p > 0.05$). In contrast, the NFV of non-Nas fluorophore decreases with the increase of $PM_{2.5}$ concentrations. This suggests that HO-HULIS is the dominant fluorophore under contaminated conditions (see Figure S14). The different degrees of increase in HO-HULIS and LO-HULIS highlight the contributions of combustion related sources and secondary sources and the increase of aerosol oxidation under high pollution levels. This also implies an increase in chromophores with aromatic or heterocyclic structures under pollution conditions, which is consistent with the indication of functional groups."*

**References:**

460     Li, P., Yue, S., Yang, X., Liu, D., Zhang, Q., Hu, W., Hou, S., Zhao, W., Ren, H., Li, G., Gao, Y., Deng, J., Xie, Q., Sun, Y., Wang, Z., and Fu, P.: Fluorescence properties and chemical composition of fine particles in the background atmosphere of North China, Advances in Atmospheric Sciences, 40, 1159-1174, https://doi.org/10.1007/s00376-022-2208-x, 2023.

**Comment #14:** *L352-353: self-contradictory statement!*

465     **Response to Comment #14:** We are sorry for the self-contradictory statement. We have revised the statement in this paragraph in the revised manuscript. Please refer to the **Response to Comment #13** for the specific modifications.

**Minor Comments:**

**Comment #15:** *L158-160: It looks odd. One can simply refer the figure and come to the discussion point.*

470     **Response to Comment #15:** We thank the reviewer for pointing this out. These sentences have been modified as follows in the revised manuscript,

*"During the observation period, the mass concentrations of carbonaceous components (i.e., OC, EC and WSOC) increase with the increase of $PM_{2.5}$ concentration (see Figure S3), and exhibit significant spatial variations across the ten sites ($p < 0.05$). As shown in Figure S1 and Table S2, the average concentrations of OC, EC and WSOC*

475     *observed at the ten sites ranged from 3.31 to 19.6 $\mu g \cdot m^{-3}$, 0.35 to 2.86 $\mu g \cdot m^{-3}$, and 1.97 to 10.6 $\mu g \cdot m^{-3}$, respectively, with the highest mean mass concentrations of OC, EC and WSOC observed in CQ, XA, and HD, respectively, while the lowest values all observed in SH. Overall, the regional average carbonaceous component concentrations show the spatial distribution trends of northwest China > southwest China > north China > east China > regional site ($p < 0.05$)."*

480     **Comment #16:** *Fig. 2: For similar Log ($MAE_{405}$), there is a considerable variability in AAE. Explain this observation.*

**Response to Comment #16:** We thank the reviewer for pointing this out. This may be due to the following reasons:

(1) The variation ranges of $MAE_{405}$ were missing in the original Figure 2. This means that the different AAE values

may not correspond to the same or similar MAE values.

485 (2) AAE value is mostly influenced by particle size, and the optical properties of light-absorbing substances and the coatings, and it may also significantly depend on the extraction method (degree of dilution) (Zhang et al., 2013). The different extraction methods used in different studies may be also a reason for the variability in AAE values.

(3) AAE is commonly used to represent the wavelength dependence of BrC light absorption within a certain

490 wavelength range, while MAE represents the light absorption capacity per unit mass of BrC, which is usually calculated by selecting a single wavelength (e.g., 365 nm, 405 nm). Therefore, AAE, which presents the light absorption characteristics of BrC within a certain wavelength range, theoretically has no fixed negative or positive relationship with MAE value calculated based on a single wavelength. In some cases, similar MAE values for BrC calculated at the same wavelength may correspond to different AAE values. For example, the

495 $MAE_{405}$ of sample A and B in this study are almost identical, but the $AAE_{300-500}$ of sample B is significantly lower than that of sample A, because the spectra of the two samples were different (see Figure R2).

[Figure]

**Figure R2.** The MAE spectra of sample A and sample B within the range of 300-500 nm. Note: The dashed line represents the position of 405 nm.

500 In the revised manuscript, the original Figure 2 (presented as new Figure 1) has been replotted by adding the variation range of $MAE_{405}$ values for the samples measured in this study, and the AAE values were recalculated within the range of 300-500 nm ($AAE_{300-500}$).

[Figure]

***Figure 1.*** *Graphical representation of optical-based BrC classes in log10(MAE405)-AAE space. The shaded regions represent very weakly light-absorbing BrC (VW-BrC), weakly light-absorbing BrC (W-BrC), moderately light-absorbing BrC (M-BrC), strongly light-absorbing BrC (S-BrC), and absorbing BC, respectively.*

**Corresponding author:** Caiqing Yan (email: cyan0325@sdu.edu.cn)

**General Comments:** *Brown carbon (BrC) is an important constituent of carbonaceous aerosols and significantly contributes to the total solar light absorption of aerosols. The manuscript titled "Optical and structural properties of atmospheric water-soluble organic carbon in China: Insights from multi-site spectroscopic measurements" presents measurements of optical properties and structural characteristics of WSOC based on different spectroscopic techniques (absorbance, fluorescence, and FTIR) from different regions of China. Overall, the study promotes a better understanding of the spatial heterogeneity of optical and structural properties of WSOC and their influencing factors (emission sources, aging processes, relative humidity (RH), etc.) in China and deepened the understanding of the contribution of WSOC fluorescence to its light absorption. However, the manuscript has many shortcomings in its current version. It needs through language editing and clarifications at many places throughout the manuscript. It also misses consistency while using different terminologies for the same parameter (for example, authors have used WSOC and WS-BrC interchangeably to refer to BrC). Yet, the study has relevance to the atmospheric research community and can be accepted for publication in the journal after major revision.*

**Response to General Comments:** We thank the reviewer for the overall supportive comments. We also appreciate the reviewer's considerable efforts in reviewing the manuscript and providing valuable comments and suggestions for the improvements and clarifications. Based on the reviewer's comments and suggestions, the manuscript is thoroughly revised. In particular, we have carefully revised the language and inappropriate and unclear expressions throughout the manuscript, so as to interpret the data more accurately and reasonably, and improve the rigor and scientific nature of the discussion. Additionally, we have taken the reviewer's suggestion to use WSOC uniformly to ensure consistency in terminologies of the same parameter in the revised manuscript. Below, we detail our responses and resulting edits to all the comments. These are organized such that we first list the review comments in italics and blue, immediately followed by our responses in normal font. To make it clear, the contents in the revised manuscript are presented in quotes and italics, while the newly added contents in the revised manuscript are underlined.

**Major Comments:**

**Comment #1:** *Methodology and elsewhere: What do you mean by "regional site (rs)" in your manuscript? Do you mean "remote/rural site"? It's confusing. Clarify.*

590   **Response to Comment #1:** We are sorry for the confusion. The "regional site" in this study does not refer to "remote/rural site". Regional sites are usually far away from urban built-up area and major pollution sources, and the distance from urban built-up areas and major pollution sources is more than 20 kilometers and less than 50 kilometers. In addition, regional site can be used as a representative receptor site for regional emissions within the region. That is, regional sites are not located in rural areas and are not as far away as remote sites.

595   In this study, the TS (1534 m a.s.l.) site, locating at the Taishan National Reference Climatological station at the summit of Mt. Tai and close to Taian City in Shandong Province, China, is less affected by anthropogenic emissions and has been widely used as a sampling site for researches on regional atmospheric pollution and atmospheric chemistry in Northern China (Chen et al., 2022; Jiang et al., 2020). Meanwhile, the HS site locates at the Atmospheric Super Monitoring Station in Jiangmen City in Guangdong Province, China and downwind of the Pearl

600   River Delta (PRD), which has been used as a representative regional receptor site for the PRD region (He et al., 2019; Xu et al., 2022). It is worth noting that the distances between the sites and the cities are around 20-30 km. In summary, TS and HS can represent the atmospheric characteristics of the North China Plain and the PRD region to some extent, respectively, so we take them as regional sites in this study.

To make it clearer, we have made the following modification in the revised manuscript,

605   *"The other two sites, Mt. Tai (TS) and Heshan (HS), are taken as regional sites in this study. TS (1534 m a.s.l.) locates at the Taishan National Reference Climatological Station at the summit of Mt. Tai and in the middle of the North China Plain, which is less affected by anthropogenic emissions and has been widely used as a sampling site for researches on regional atmospheric pollution and atmospheric chemistry in Northern China (Chen et al., 2022; Jiang et al., 2020). The HS site locates at the Atmospheric Environmental Monitoring Super-station in Guangdong,*

610   *China and downwind of the Pearl River Delta (PRD), which is mainly surrounded by farmland protection areas and forest land with no obvious industrial or urban traffic pollution sources in the vicinity, and has been used as a representative regional receptor site for the PRD region (He et al., 2019; Xu et al., 2022)."*

630    **Comment #2:** *Methodology (section 2.3): Equation S9 in text S3 is incorrect (could be a typo error). The equation should consist both mass scattering efficiency (MSE) and MAE. Recheck and correct it.*

**Response to Comment #2:** We thank the reviewer for the careful editing. We have rechecked and corrected the equation S9 as follows,

$$\frac{dSFE}{d\lambda} = -\frac{1}{4} \frac{dS(\lambda)}{d\lambda} \ \tau_{atm}^2(\lambda) \ (1 - F_c) \left[ 2(1 - \alpha_s)^2 \ \beta(\lambda) \ MSE(\lambda) - 4\alpha_s \ MAE(\lambda) \right] \qquad (S9)$$

635    *"where $dS(\lambda)/d\lambda$ is the solar irradiance ($W \cdot m^{-2} \cdot nm^{-1}$) obtained from the ASTM G173-03 reference spectra, $\tau_{atm}$ is the atmospheric transmission (0.79), Fc is the cloud fraction (0.6), $\alpha_s$ is the surface albedo (average 0.19), $\beta$ is the backscatter fraction, MSE and MAE are the mass scattering efficiency and mass absorption efficiency of WSOC, respectively. It should be noted that $\beta = 0$ and only light absorption is considered in the calculation in this study."* (in the Supplement)

640    **Comment #3:** *Methodology (section 2.3, text S4) and section 3.2 of results section: The author measured WSOC absorbance from 250-700 nm and used WSOC absorbance from 250 to 400 nm in PMF model for source apportionment. However, it is well known that WSOC absorbance < 340 nm is highly influenced by absorbance*

*from nitrate aerosols. Did authors consider this aspect during PMF run? How this will impact the findings?*

**Response to Comment #3:** We thank the reviewer for raising this good question. We have checked the light absorption spectra of nitrate and nitrite solutions reported in previous studies and found that the absorption of nitrate is mainly concentrated below 250 nm (Ai, 2019; Afsana et al., 2022; Dong et al., 2022; Li et al., 2016; Pons et al., 2017; Zhang et al., 2021; Zhang et al., 2023) (see Figure R1). That means, the absorbance interference of nitrate aerosols should be mainly below 250 nm. In this study, WSOC light absorbance from 250 to 500 nm has been used as input in PMF model in the revised manuscript, and the results indicate that the absorption spectra of all light-absorbing factors resolved by PMF are close to those of previously identified BrC light-absorbing species. Therefore, the impacts by light absorption of nitrate may be negligible on the PMF results.

[Figure]

**Figure R1.** The light absorption spectra of nitrates measured in different studies.

675  **Comment #4:** *Results and Discussion (section 3.1): The authors observed a significant spatial variability in WSOC, OC, EC, etc. across ten sites. What could be the potential reasons (e.g., different sources, metrology, etc.) behind this variability, discuss briefly?*

**Response to Comment #4:** We thank the reviewer for the constructive suggestion. In the revised manuscript, we have revised this section and briefly discussed the possible reasons for the spatial variation of carbonaceous
680  components as follows,

[revised manuscript text omitted]

**Comment #5:** *Results and Discussion (section 3.1): The authors compared $Abs_{365}$ and $MAE_{365}$ values between different regions (e.g., northwest China, southwest China, etc.). Did you carry out any significance test to check whether difference was significant or not?*

**Response to Comment #5:** We thank the reviewer for pointing this out. We have taken the reviewer's suggestion

745      and carried out a *t-test* (two-sample heteroskedasticity assumption) to check the significant differences in $Abs_{365}$ and $MAE_{365}$ in these regions, and made the following modifications in the revised manuscript,

     *"As shown in* Table 1 and Figure S1, the average $Abs_{365}$ (1.12 ± 0.53-13.1 ± 6.95 $Mm^{-1}$) and $MAE_{365}$ (0.56 ± 0.11-

     1.26 ± 0.34 $m^2 \cdot g^{-1}$) of WSOC at the ten sites display significant spatial discrepancies (p < 0.05), with HD (SH) has

     the highest (lowest) average $Abs_{365}$ (13.1 ± 6.95 $Mm^{-1}$ (1.12 ± 0.53 $Mm^{-1}$)) and $MAE_{365}$ (1.26 ± 0.34 $m^2 \cdot g^{-1}$ (0.56 ±

750      0.11 $m^2 \cdot g^{-1}$)), respectively. *$MAE_{365}$ in SH, CD, TS and HS (0.56 ± 0.11-0.74 ± 0.24 $m^2 \cdot g^{-1}$ on average) are comparable to those reported in light-polluted areas such as in Guangzhou, Lulang, Waliguan, Urumqi in China and Los Angeles in the USA (0.48-0.81 $m^2 \cdot g^{-1}$) (Fan et al., 2016; Liu et al., 2018; Soleimanian et al., 2020; Wu et al., 2020; Xu et al., 2020; Zhong et al., 2023). However, they are all lower than those in TJ, HD, QD, NJ, XA and CQ (0.89 ± 0.22-1.26 ± 0.34 $m^2 \cdot g^{-1}$ on average), which are comparable to those reported in heavy pollution areas such as in*

755      *Beijing, Xining, Yinchuan, Lanzhou, Taipei in China, and Patiala and Mohanpur in India (0.93-1.30 $m^2 \cdot g^{-1}$) (Cheng et al., 2016; Srinivas et al., 2016; Dey et al., 2021; Zhong et al., 2023; Ting et al., 2022). In this study,* the light absorption of WSOC in different region is significantly different (p < 0.05), with *the regional average $Abs_{365}$ and $MAE_{365}$* displaying *as northwest China > southwest China > north China > east China > regional site. Moreover, the average $Abs_{365}$ and $MAE_{365}$ are higher in Northern China (including TJ, HD, QD, and XA, 7.34 ± 5.21 $Mm^{-1}$*

760      *and 1.02 ± 0.29 $m^2 \cdot g^{-1}$, respectively) than in Southern China (including NJ, SH, CD, and CQ, 5.86 ± 3.91 $Mm^{-1}$ and 0.78 ± 0.23 $m^2 g^{-1}$, respectively) and regional sites (e.g., TS and HS, 2.91 ± 1.38 $Mm^{-1}$ and 0.72 ± 0.23 $m^2 \cdot g^{-1}$, respectively)* (p < 0.01)*, and higher in inland (8.24 ± 4.75 $Mm^{-1}$ and 0.91 ± 0.27 $m^2 \cdot g^{-1}$, respectively) than in coastal areas (4.37 ± 3.52 $Mm^{-1}$ and 0.88 ± 0.32 $m^2 \cdot g^{-1}$, respectively)* (p < 0.01, see Figure S5)*, which are consistent with the regional differences in carbonaceous component mass concentrations."*

765      **Comment #6:** *Results and Discussion (section 3.1): The sampling durations were different at different sites representing different administrative regions (Table S1). Do you think the "day versus night variability" in optical properties could have also contributed to the inter-regional variability observed in optical properties of WS-BrC in your study.*

     **Response to Comment #6:** Yes, we agree with the reviewer that "day versus night variability" in optical properties

770      may contribute to inter-regional variability. However, it is not the case in this study and the effect of day versus night variability can be ignored in this study. The specific reasons are as follows:

(1) During the sampling period of this study, daytime and nighttime samples were only collected at three sites, including HD, NJ and TS. We compare $Abs_{365}$ and $MAE_{365}$ of daytime and nighttime WSOC at the three sites

and find that there is a small difference in light absorption between the daytime and nighttime WSOC (see
775 Figure R2). Furthermore, the significance test of the daytime and nighttime data shows that the difference is
not significant ($p > 0.05$, *t-test*).

[Figure]

**Figure R2.** The average $Abs_{365}$ and $MAE_{365}$ values during daytime and nighttime at HD, NJ and TS.

(2) Since daytime and nighttime samples were not collected at all sites, in order to maintain consistency, we
780 calculated the daily average values through daytime and nighttime samples during data processing, and then
compared the differences of daily mean values among the ten sites, and did not compare the daytime and
nighttime data in this study. Therefore, in the revised manuscript, the following sentence has been added.

*"It is worth noting that in this study, the daily average of the parameters measured at each site is used for
subsequent summary and comparison."*

785 **Comment #7:** *Lines 208-210: The authors reported that light absorbing ability (SFE) of WS-BrC and mass
concentration of WSOC are directly proportional (related). How did authors come with such conclusions? Please
cite relevant studies in this context.*

**Response to Comment #7:** Thanks for the reviewer's suggestion. In the revised manuscript, we have revised the
statement and cited relevant references. The related paragraph has been revised as follows,

[revised manuscript text omitted]

**Comment #8:** *Fig. 6c-d: Is this the integrated absorbance from 250-400 nm, clarify?*

**Response to Comment #8:** We are sorry for the confusion. In the original draft, we calculated the averaged light
840  absorption contribution with light absorbance at each wavelength. In the revised manuscript, we recalculated the average contribution based on the integrated absorbance from 250-500 nm of each factor.

For clarity, we have revised the title of the new Figure 5 (original Figure 6) in the revised manuscript as follows,

*"**Figure 5.** The average light absorption spectra of the absorption factors resolved by PMF model at (a) East China sites (unimodal absorption spectral type) and (b) outside East China sites (bimodal absorption spectral type),*
845  *as well as the average contribution by each factor calculated according to the integral absorbance from 250-500 nm at both types of sites (panel c and d)."*

**Comment #9:** *Line 341-346: Please revisit this portion, especially, portion where authors mentioned that photochemical bleaching will be higher during severe pollution days. In fact, opposite is likely to be true as lower pollution levels mean higher visibility, resulting in higher availability of solar flux.*

850  **Response to Comment #9:** We thank the reviewer for pointing this out. In the revised manuscript, this paragraph has been revised as follows,

*"To investigate the influence of air quality levels on the light absorption properties of WSOC, the sampling days are classified into five pollution levels including clean (0-35 µg·m⁻³), relatively clean (35-75 µg·m⁻³), slightly polluted (75-115 µg·m⁻³), moderately polluted (115-150 µg·m⁻³), and heavily polluted (> 150 µg·m⁻³) according to the national ambient air quality daily Grade-II standard threshold values and ambient air quality indices. As shown in Figure 7a and b, $Abs_{365}$ and $MAE_{365}$ of WSOC both increase with the increase of pollution levels, in which $Abs_{365}$ changes significantly ($p < 0.01$) while $MAE_{365}$ changes relatively gently. The enhancement of WSOC light absorption under high pollution conditions may be related to the increase of WSOC concentration, light absorption capacity and light-absorbing species. Previous studies have reported that the mass fractions of oxidized organic aerosols increase significantly with the increase of $PM_{2.5}$ mass concentration, and the oxidized organic aerosols contain a large number of light-absorbing species such as nitroaromatics compounds (You et al., 2024). In this study, the relative abundances of O-H, C=C and $R-ONO_2$ functional groups, which are related to aromatic compounds and have a good positive correlation with the light absorption of WSOC, increase with $PM_{2.5}$ mass concentration (see Figure S14, and discussion in the next section). Additionally, the accumulation of anthropogenic emissions (especially those sources with strong light-absorbing BrC such as biomass burning and coal combustion sources) at high pollution levels will lead to an increase in BrC chromophore types and overall light absorption capacity (Li et al., 2020a; Tang et al., 2020; Wei et al., 2020)."*

[Figure]

***Figure S14.*** *Variations of the relative abundance of functional groups (a) C=C, (b) O-H and (c) $R-ONO_2$ and the proportion of (d) LO-HULIS, (e) HO-HULIS and (f) non-Nas fluorophores with $PM_{2.5}$ mass concentrations.*

Accordingly, the $Abs_{365, WISOC}$ should be around 0.62 to 1.47 times of $Abs_{365, WSOC}$ among the ten sites. As shown in Table R1, the $Abs_{365, WISOC}$ at some sites (e.g., TJ, HD, QD, XA, CQ and TS) are higher than $Abs_{365, WSOC}$, which should be given special attention, while $Abs_{365, WSOC}$ is much stronger at other sites (e.g., NJ, SH, CD and HS). *However*, it is worth mentioning that among the ten sites, only the difference in light absorption between WSOC and WISOC in QD and CD is significant ($p < 0.05$, *t-test*), with the $Abs_{365, WISOC}$ at the two sites being 1.47 and 0.62 times that of $Abs_{365, WSOC}$, respectively.

**Table R1**. The mass concentrations and light absorption coefficients of WSOC and WISOC in this study.

| Sites | WISOC/WSOC | WISOC/OC (%) | $Abs_{365, WSOC}$ (Mm$^{-1}$) | $Abs_{365, WISOC}$ (Mm$^{-1}$) | $Abs_{365, WISOC}/Abs_{365, WSOC}$ |
|---|---|---|---|---|---|
| Tianjin (TJ) | 0.74±0.36 | 40.1±11.8 | 5.57±3.83 | 7.08±7.15 | 1.18±0.58 |
| Handan (HD) | 0.90±0.41 | 44.8±13.3 | 13.1±6.95 | 18.3±7.57 | 1.45±0.65 |
| Qingdao (QD) | 0.92±0.55 | 44.9±11.6 | 4.80±3.09 | 7.31±5.30 | 1.47±0.89 |
| Nanjing (NJ) | 0.59±0.31 | 34.8±11.9 | 6.26±3.26 | 6.72±6.66 | 0.94±0.50 |
| Shanghai (SH) | 0.66±0.19 | 39.0±6.99 | 1.12±0.53 | 1.16±0.66 | 1.06±0.30 |
| Xi'an (XA) | 0.77±0.23 | 42.7±6.93 | 10.6±4.42 | 12.6±5.56 | 1.24±0.36 |
| Chengdu (CD) | 0.39±0.22 | 26.7±10.0 | 5.99±2.61 | 3.67±2.27 | 0.62±0.35 |
| Chongqing (CQ) | 0.89±0.18 | 46.6±4.94 | 10.6±4.10 | 15.0±6.15 | 1.42±0.29 |
| Mt. Tai (TS) | 0.73±0.69 | 34.9±19.6 | 2.66±1.22 | 3.51±3.95 | 1.16±1.11 |
| Heshan (HS) | 0.61±0.30 | 35.6±11.9 | 3.76±1.55 | 3.86±2.65 | 0.97±0.47 |

Taken together, although $MAE_{365, WISOC}$ and $Abs_{365, WISOC}$ may be stronger compared to $MAE_{365, WSOC}$ and $Abs_{365, WSOC}$, the estimated light absorption by WSOC and WISOC at the most sites during the study period show insignificant difference ($p > 0.05$, t-test). Relatively, the results of this study suggest that the light absorption contribution of WISOC in winter in northern cities is more noteworthy.

(3) Finally, the findings of this study show that combining multiple spectral results (light absorption spectrum, fluorescence spectrum and infrared spectrum) can provide a deeper understanding of the light-absorbing characteristics of WSOC. This study can also provide reference for further study of WISOC light absorption properties and its structural composition.

(4) Therefore, we have added a description of water-soluble and water-insoluble BrC in the revised manuscript, and added the limitations that this study only focuses on water-soluble BrC in the conclusion sections as follows,

① In the introduction section:

*"Solvent-soluble organic carbon (e.g., water-soluble organic carbon, WSOC; methanol-soluble organic carbon, MSOC) is often used to act as a substitute of BrC. In particular, light absorption of WSOC has been extensively studied, due to its widespread presence and high atmospheric abundance in the atmosphere, as well as mature extraction methods, although some previous studies have indicated that water-insoluble OC (WISOC) contains more light-absorbing BrC (Cao et al., 2021; Chen et al., 2024b; Cheng et al., 2016; Yan et al., 2017). Absorption and fluorescence spectroscopy are two of the most widely used methods to reveal optical properties of WSOC (Wang et al., 2022b; Wu et al., 2021)."*

*"In this study, $PM_{2.5}$ is collected from ten sites in different regions of China. The mass concentration, light absorption and fluorescent spectra, and functional group structures of WSOC (a substitute of water-soluble BrC) at the ten sites are analyzed using a unified method. The objectives of this study include: …Furthermore, this study can also provide reference for the future study of the light absorption properties and structural composition of WISOC."*

② In the conclusion section:

*"Additionally, it is important to note that this study only focused on WSOC. Since the WISOC may have stronger light absorption capacity, further research on light absorption, composition and structure of WISOC (especially in northern China in winter) and the correlation between them are also needed in the future."*

**Comment #16:** *Line 61: Should be "Spectroscopy-based studies conducted…"*

**Response to Comment #16:** We thank the reviewer's careful editing and kind reminder. The sentence has been revised accordingly,

*"Spectroscopy-based studies conducted in different countries or regions all over the world show that there are significant spatiotemporal differences in the light absorption characteristics of WSOC."*

**Comment #17:** *Line 81: "methods are often used separately in previous studies"?*

**Response to Comment #17:** We apologize for the confusion. We want to express that the results of different spectral methods (e.g., absorption spectra, fluorescence spectra and FTIR spectra) were often discussed separately in previous studies, without delving into the relationship between these spectral results. Exploring the relationship between different spectra is one of the objectives of this study. To make it more accurate, we have revised this sentence as follows,

*"However, in previous spectroscopy-based studies, the results of different spectral methods are often discussed separately, without in-depth discussion of the relationship between different spectra, which limits the full and comprehensive prediction of the optical and structural characteristics of BrC."*

1030     **Comment #18:** *Line 82-86: Difficult to follow as this is a very long sentence. I suggest to break it into smaller sentences.*

**Response to Comment #18:** We appreciate the reviewer's kind reminder and apologize for the bad feeling. We have taken the reviewer's suggestion and broken the long sentence into smaller ones.

*"In this study, PM$_{2.5}$ is collected from ten sites in different regions of China. The mass concentration, light*

1035     *absorption and fluorescent spectra, and functional group structures of WSOC (a substitute of water-soluble BrC) at the ten sites are analyzed using a unified method. The objectives of this study include: (1) to explore the spatial heterogeneity of optical properties of WSOC in different regions of China and its influencing factors, (2) to reveal the relationship between light absorption, fluorescence and functional group structure of WSOC, and (3) to quantify the contribution of fluorescent chromophores to the light absorption of WSOC on the basis of multi-site spectral*

1040     *datasets."*

**Comment #19:** *Line 110: should be "0.45 μm pore-size PTFE syringe filter".*

**Response to Comment #19:** Thanks for the reviewer's kind reminder. The sentence has been revised accordingly.

*"A portion of each filter (about 6 cm$^2$) is extracted with ultrapure water (>18.2 MΩ·cm, 25 °C, Direct-Q, Millipore) by ultra-sonication for 30 min, and then the extract is filtered through a 0.45 μm pore-size PTFE syringe filter (Pall,*

1045     *USA) to remove water-insoluble materials."*

**Comment #20:** *Line 118-120: Break it into two sentences like "…:calculated. More details can be found in Text S2."*

**Response to Comment #20:** We take the reviewer's suggestion and break it into two sentences accordingly.

*"Light absorption parameters such as light absorption coefficients at 365 nm (Abs$_{365}$), mass absorption efficiency*

1050     *at 365 nm and 405 nm (MAE$_{365}$ and MAE$_{405}$), Ångström exponent over 300-500 nm (AAE$_{300-500}$) are calculated. More details on the calculation methods can be found in Text S2."*

**Comment #21:** *Line 152: "Additionally, the XGBoost model is also used to…" "also" is redundant in this sentence.*

**Response to Comment #21:** Thanks for the kind reminder. "also" has been removed from this sentence accordingly, and this sentence has been revised to,

1055     *"Additionally, the XGBoost model is used to evaluate the influence of conventional gas parameters (e.g., CO, SO₂,*

    *O₃, NO₂) on the light absorption of WSOC."*

**Comment #22:** *Line 169: "···mass concentrations of carbonaceous components at HS site are not that low···"*
*Compared to what, clarify?*

**Response to Comment #22:** We are sorry for the confusion. The sentence has been revised as follows,

1060     *"Furthermore, it is worth noting that the regional site TS in NCP has a relatively low mass concentration of*

    *carbonaceous components compared to urban sites, which may be due to its high altitude (~1500 m) and low local*

    *anthropogenic activities (Jiang et al., 2020). In contrast, the mass concentrations of carbonaceous components at*

    *HS (another regional site) in the PRD region are relatively higher compared to TS site. The backward air mass*

    *trajectory analysis indicates that more than 80% of the air masses arriving at the HS site originated from the PRD*

1065     *region and are accompanied by low wind speeds (1.54 m·s⁻¹ on average during the sampling period). This suggests*

    *that there may be high anthropogenic emissions in the PRD region during the winter sampling period."*

*significant spatial variation across the ten sites (p < 0.05) (see Figure 1 and Table S2)." Which test did you use to*
*derive significance level?*

1075     **Response to Comment #23:** Thanks for the good question. In this study, a two-sample t-test under heteroscedasticity

with the 95% confidence level was used to evaluate the significance level of data differences. In the section *"2.6*

*Relationship and influencing factor analysis"* of the revised manuscript, we have added the following explanation

of the test method used in this study.

    *"The calculation is mainly carried out through SPSS software (IBM SPSS Statistics 23). Notably, t-test (two-sample*

1080     *testing under heteroscedasticity, at the 95% confidence level) is conducted to evaluate the significance level of data*

    *differences in this study."*

**Comment #24:** *Line 308-309: The sentence is confusing. Rewrite it.*

**Response to Comment #24:** We are sorry for the confusion. The sentence has been revised as follows,

*"The contributions by different light absorption factors vary significantly at different wavelengths (see Figure S12)."*

1085 **Comment #25:** *Line 341: Change "great" to "large".*

**Response to Comment #25:** Thanks. The relevant sentence has been deleted in the revised manuscript.

**Comment #26:** *Line 378: Typo? "WOSC" should be "WSOC"*

**Response to Comment #26:** We apologize for the typo and thank the reviewer for the correction. "WOSC" has been replaced by "WSOC" in the revised manuscript.

1090 *"...indicating that the fluorescent components may contribute to WSOC light absorption to some extent."*

**Comment #27:** *"In contrast, the relationships between $MAE_{365}$ values and functional groups may differ from $Abs_{365}$. Similarly, C=C, O-H and R-ONO$_2$ exhibit the strongest correlations with $MAE_{365}$ at most sites (e.g., TJ, QD, SH, TS, and HS)." Similar to what? These sentences are confusing. Rewrite them.*

**Response to Comment #27:** We apologize for the confusion. The sentence has been rewritten as follows,

1095 *"In contrast, the relationships of functional groups with $MAE_{365}$ are slightly different from that with $Abs_{365}$. Overall, C=C, O-H and R-ONO$_2$ exhibit the strongest correlations with $MAE_{365}$ at most sites (e.g., TJ, QD, SH, TS, and HS). C=O and C-O, which may be related to carboxylic acids, phenols and esters, show positive correlations with $MAE_{365}$ at some sites (e.g., HD, NJ, XA and CQ)."*

**Comment #28:** *Line 413: "discrepancies" doesn't seem to be the write word here. Replace it.*

1100 **Response to Comment #28:** We thank the reviewer's kind reminder. The "discrepancies" has been replaced by "spatial variations", and the sentence has been revised as follows,

*"Based on the same measurement methods and data processing processes, light absorption, fluorescence and FTIR spectra analysis are combined to investigate the optical properties and functional group characteristics of WSOC at ten sites in different regions of China. The spatial variations at various sites and the relationships between*

1105    *absorbance, fluorescence, and functional groups of WSOC are revealed."*